# Hoyer Regularizer is all you need for Ultra Low-Latency Spiking Neural Networks

## Abstract

Spiking Neural networks (SNN) have emerged as an attractive spatio-temporal computing paradigm for a wide range of low-power vision tasks. However, state-of-the-art (SOTA) SNN models either incur multiple time steps which hinder their deployment in real-time use cases or increase the training complexity significantly. To mitigate this concern, we present a training framework (from scratch) for one-time-step SNNs that uses a novel variant of the recently proposed Hoyer regularizer. We estimate the threshold of each SNN layer as the Hoyer extremum of a clipped version of its activation map, where the clipping threshold is trained using gradient descent with our Hoyer regularizer. This approach not only downscales the value of the trainable threshold, thereby emitting a large number of spikes for weight update with a limited number of iterations (due to only one time step) but also shifts the membrane potential values away from the threshold, thereby mitigating the effect of noise that can degrade the SNN accuracy. Our approach outperforms existing spiking, binary, and adder neural networks in terms of the accuracy-FLOPs trade-off for complex image recognition tasks. Downstream experiments on object detection also demonstrate the efficacy of our approach. Codes will be made publicly available.

## 1 Introduction & Related Works

Due to its high activation sparsity and use of cheaper accumulates (AC) instead of energy-expensive multiply-and-accumulates (MAC), SNNs have emerged as a promising low-power alternative to compute- and memory-expensive deep neural networks (DNN) (Indiveri et al., 2011; Pfeiffer et al., 2018; Cao et al., 2015). Because SNNs receive and transmit information via spikes, analog inputs have to be encoded with a sequence of spikes using techniques such as rate coding (Diehl et al., 2016), temporal coding (Comsa et al., 2020), direct encoding (Rathi et al., 2020a) and rank-order coding (Kheradpisheh et al., 2018). In addition to accommodating various forms of spike encoding, supervised training algorithms for SNNs have overcome various roadblocks associated with the discontinuous spike activation function (Lee et al., 2016; Kim et al., 2020). Moreover, previous SNN efforts propose batch normalization (BN) techniques (Kim et al., 2020; Zheng et al., 2021) that leverage the temporal dynamics with rate/direct encoding. However, most of these efforts require multiple time steps which increases training and inference costs compared to non-spiking counterparts for static vision tasks. The training effort is high because backpropagation must integrate the gradients over an SNN that is unrolled once for each time step (Panda et al., 2020). Moreover, the multiple forward passes result in an increased number of spikes, which degrades the SNN's energy efficiency, both during training and inference, and possibly offsets the compute advantage of the ACs. The multiple time steps also increase the inference complexity because of the need for input encoding logic and the increased latency associated with requiring one forward pass per time step. To mitigate these concerns, we propose one-time-step SNNs that do not require any non-spiking DNN pre-training and are more compute-efficient than existing multi-time-step SNNs. Without any temporal overhead, these SNNs are similar to vanilla feed-forward DNNs, with Heaviside activation functions (McCulloch & Pitts, 1943). These SNNs are also similar to sparsity-induced or uni-polar binary neural networks (BNNs) (Wang et al., 2020b) that have 0 and 1 as two states. However, these BNNs do not yield SOTA accuracy like the bi-polar BNNs (Diffenderfer & Kailkhura, 2021) that has 1 and -1 as two states. A recent SNN work (Chowdhury et al., 2021) also proposed the use of one time-step, however, it required CNN pre-training, followed by iterative SNN training from 5 to 1 steps, significantly increasing the training complexity, particularly for ImageNet-level tasks. Note

that there have been significant efforts in the SNN community to reduce the number of time steps via optimal DNN-to-SNN conversion (Bu et al., 2022b; Deng et al., 2021), lottery ticket hypothesis (Kim et al., 2022c), and neural architecture search (Kim et al., 2022b). However, none of these techniques have been shown to train one-time-step SNNs without significant accuracy loss.

**Our Contributions**. Our training framework is based on a novel application of the Hoyer regularizer and a novel Hoyer spike layer. More specifically, our spike layer threshold is training-input-dependent and is set to the Hoyer extremum of a clipped version of the membrane potential tensor, where the clipping threshold (existing SNNs use this as the threshold) is trained using gradient descent with our Hoyer regularizer. In this way, compared to SOTA one-time-step non-iteratively trained SNNs, our threshold increases the rate of weight updates and our Hoyer regularizer shifts the membrane potential distribution away from this threshold, improving convergence.

We consistently surpass the accuracies obtained by SOTA one-time-step SNNs (Chowdhury et al., 2021) on diverse image recognition datasets with different convolutional architectures, while reducing the average training time by $\sim19\times$. Compared to binary neural networks (BNN) and adder neural network (AddNN) models, our SNN models yield similar test accuracy with a $\sim5.5\times$ reduction in the floating point operations (FLOPs) count, thanks to the extreme sparsity enabled by our training framework. Downstream tasks on object detection also demonstrate that our approach surpasses the test mAP of existing BNNs and SNNs.

## 2 PRELIMINARIES ON HOYER REGULARIZERS

Based on the interplay between L1 and L2 norms, a new measure of sparsity was first introduced in (Hoyer, 2004), based on which reference (Yang et al., 2020) proposed a new regularizer, termed the Hoyer regularizer for the trainable weights that was incorporated into the loss term to train DNNs. We adopt the same form of Hoyer regularizer for the membrane potential to train our SNN models as $H(\boldsymbol{u}_l) = \left( \frac{\|\boldsymbol{u}_l\|_1}{\|\boldsymbol{u}_l\|_2} \right)^2$ (Kurtz et al., 2020). Here, $\|\boldsymbol{u}_l\|_i$ represents the Li norm of the tensor $\boldsymbol{u}_l$, and the superscript $t$ for the time step is omitted for simplicity. Compared to the L1 and L2 regularizers, the Hoyer regularizer has scale-invariance (similar to the L0 regularizer). It is also differentiable almost everywhere, as shown in equation 1, where $|\boldsymbol{u}_l|$ represents the element-wise absolute of the tensor $\boldsymbol{u}_l$.

$$\frac{\partial H(\boldsymbol{u}_l)}{\partial \boldsymbol{u}_l} = 2sign(\boldsymbol{u}_l)\frac{\|\boldsymbol{u}_l\|_1}{\|\boldsymbol{u}_l\|_2^4}(\|\boldsymbol{u}_l\|_2^2 - \|\boldsymbol{u}_l\|_1|\boldsymbol{u}_l|) \tag{1}$$

Letting the gradient $\frac{\partial H(\boldsymbol{u}_l)}{\partial \boldsymbol{u}_l} = 0$, we estimate the value of the Hoyer extremum as $Ext(\boldsymbol{u}_l) = \frac{\|\boldsymbol{u}_l\|_2^2}{\|\boldsymbol{u}_l\|_1}$.

This extremum is actually the minimum, because the second derivative is greater than zero for any value of the output element. Training with the Hoyer regularizer can effectively help push the activation values that are larger than the extremum ($\boldsymbol{u}_l > Ext(\boldsymbol{u}_l)$) even larger and those that are smaller than the extremum ($\boldsymbol{u}_l < Ext(\boldsymbol{u}_l)$) even smaller.

## 3 PROPOSED TRAINING FRAMEWORK

Our approach is inspired by the fact that Hoyer regularizers can shift the pre-activation distributions away from the Hoyer extremum in a non-spiking DNN (Yang et al., 2020). Our principal insight is that setting the SNN threshold to this extremum shifts the distribution of the membrane potentials away from the threshold value, reducing noise and thereby improving convergence. To achieve this goal for one-time-step SNNs we present a novel *Hoyer spike layer* that sets the threshold based upon a *Hoyer regularized training process*, as described below.

### 3.1 HOYER SPIKE LAYER

In this work, we adopt a time-independent variant of the popular Leaky Integrate and Fire (LIF) representation, as illustrated in Eq. 2, to model the spiking neuron with one time-step.

$$\boldsymbol{u}_l = \boldsymbol{w}_l\boldsymbol{o}_{l-1} \quad \boldsymbol{z}_l = \frac{\boldsymbol{u}_l}{v_l^{th}} \quad \boldsymbol{o}_l = \begin{cases} 1, & \text{if } \boldsymbol{z}_l \geq 1; \\ 0, & \text{otherwise} \end{cases} \tag{2}$$

where $\boldsymbol{z}_l$ denotes the normalized membrane potential. Such a neuron model with a unit step activation function is difficult to optimize even with the recently proposed surrogate gradient descent

techniques for multi-time-step SNNs (Panda et al., 2020; Panda & Roy, 2016), which either approximates the spiking neuron functionality with a continuous differentiable model or uses surrogate gradients to approximate the real gradients. This is because the average number of spikes with only one time step is too low to adjust the weights sufficiently using gradient descent with only one iteration available per input. This is because if a pre-synaptic neuron does not emit a spike, the synaptic weight connected to it cannot be updated because its gradient from neuron $i$ to $j$ is calculated as $g_{u_j} \times o_i$, where $g_{u_j}$ is the gradient of the membrane potential $u_j$ and $o_i$ is the output of the neuron $i$ Therefore, it is crucial to reduce the value of the threshold to generate enough spikes for better network convergence. Note that a sufficiently low value of threshold can generate a spike for every neuron, but that would yield random outputs in the final classifier layer.

Previous works (Datta et al., 2021; Rathi et al., 2020a) show that the number of SNN time steps can be reduced by training the threshold term $v_l^{th}$ using gradient descent. However, our experiments indicate that, for one-time-step SNNs, this approach still yields thresholds that produce significant drops in accuracy. In contrast, we propose to dynamically down-scale the threshold (see Fig. 1(a)) based on the membrane potential tensor using our proposed form of the Hoyer regularizer. In particular, we clip the membrane potential

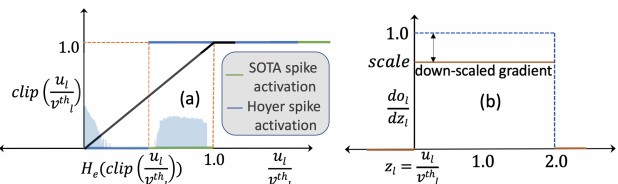

Figure 1: (a) Comparison of our Hoyer spike activation function with existing activation functions where the blue distribution denotes the shifting of the membrane potential away from the threshold using Hoyer regularized training, (b) Proposed derivative of our Hoyer activation function.

tensor corresponding to each convolutional layer to the trainable threshold $v_l^{th}$ obtained from the gradient descent with our Hoyer loss, as detailed later in Eq. 11. Unlike existing approaches (Datta & Beerel, 2022; Rathi et al., 2020a) that require $v_l^{th}$ to be initialized from a pre-trained non-spiking model, our approach can be used to train SNNs from scratch with a Kaiming uniform initialization (He et al., 2015) for both the weights and thresholds. In particular, the down-scaled threshold value for each layer is computed as the Hoyer extremum of the clipped membrane potential tensor, as shown in Fig. 1(a) and mathematically defined as follows.

$$z_l^{clip} = \begin{cases} 1, & \text{if } \mathbf{z}_l > 1 \\ \mathbf{z}_l, & \text{if } 0 \leq \mathbf{z}_l \leq 1 \\ 0, & \text{if } \mathbf{z}_l < 0 \end{cases} \qquad \mathbf{o}_l = h_s(\mathbf{z}_l) = \begin{cases} 1, & \text{if } \mathbf{z}_l \geq Ext(\mathbf{z}_l^{clip}); \\ 0, & \text{otherwise} \end{cases} \tag{3}$$

Note that our threshold $Ext(\mathbf{z}_l^{clip})$ is indeed less than the trainable threshold $v_l^{th}$ used in earlier works (Datta & Beerel, 2022; Rathi et al., 2020a) for any output, and the proof is shown in Appendix A.1. Moreover, we observe that the Hoyer extremum in each layer changes only slightly during the later stages of training, which indicates that it is most likely an inherent attribute of the dataset and model architecture. Hence, to estimate the threshold during inference, we calculate the exponential average of the Hoyer extremums during training (similar to BN), and use the same during inference.

## 3.2 Hoyer Regularized Training

The loss function ($L_{total}$) of our proposed approach is shown below in Eq. 4.

$$L_{total} = L_{CE} + L_H = L_{CE} + \lambda_H \sum_{l=1}^{L-1} H(\mathbf{u}_l) \tag{4}$$

where $L_{CE}$ denotes the cross-entropy loss calculated on the softmax output of the last layer $L$, and $L_H$ represents the Hoyer regularizer calculated on the output of each convolutional and fully-connected layer, except the last layer. The weight update for the last layer is computed as

$$\Delta W_L = \frac{\partial L_{CE}}{\partial \mathbf{w}_L} + \lambda_H \frac{\partial L_H}{\partial \mathbf{w}_L} = \frac{\partial L_{CE}}{\partial \mathbf{u}_L}\frac{\partial \mathbf{u}_L}{\partial \mathbf{w}_L} + \lambda_H \frac{\partial L_H}{\partial \mathbf{u}_L}\frac{\partial \mathbf{u}_L}{\partial \mathbf{w}_L} = (\mathbf{s} - \mathbf{y})\mathbf{o}_{L-1} + \lambda_H \frac{\partial H(\mathbf{u}_L)}{\partial \mathbf{u}_L}\mathbf{o}_{L-1} \tag{5}$$

$$\frac{\partial L_{CE}}{\partial \mathbf{o}_{L-1}} = \frac{\partial L_{CE}}{\partial \mathbf{u}_L}\frac{\partial \mathbf{u}_L}{\partial \mathbf{o}_{L-1}} = (\mathbf{s} - \mathbf{y})\mathbf{w}_L \tag{6}$$

where $\boldsymbol{s}$ denotes the output softmax tensor, i.e., $s_i = \frac{e^{u_L^i}}{\sum_{k=1}^N u_L^k}$ where $u_L^i$ and $u_L^k$ denote the $i^{th}$ and $k^{th}$ elements of the membrane potential of the last layer $L$, and $N$ denotes the number of classes. Note that $\boldsymbol{y}$ denotes the one-hot encoded tensor of the true label, and $\frac{\partial H(\boldsymbol{u}_L)}{\partial \boldsymbol{u}_L}$ is computed using Eq. 1. The last layer does not have any threshold and hence does not emit any spike.

For a hidden layer $l$, the weight update is computed as

$$\Delta W_l = \frac{\partial L_{CE}}{\partial \boldsymbol{w}_l} + \lambda_H \frac{\partial L_H}{\partial \boldsymbol{w}_l} = \frac{\partial L_{CE}}{\partial \boldsymbol{o}_l}\frac{\partial \boldsymbol{z}_l}{\partial \boldsymbol{u}_l}\frac{\partial \boldsymbol{u}_l}{\partial \boldsymbol{w}_l} + \lambda_H \frac{\partial L_H}{\partial \boldsymbol{u}_l}\frac{\partial \boldsymbol{u}_l}{\partial \boldsymbol{w}_l} = \frac{\partial L_{CE}}{\partial \boldsymbol{o}_l}\frac{\partial \boldsymbol{o}_l}{\partial \boldsymbol{z}_l}\frac{\boldsymbol{o}_{l-1}}{v_l^{th}} + \lambda_H \frac{\partial L_H}{\partial \boldsymbol{u}_l}\boldsymbol{o}_{l-1} \tag{7}$$

where $\frac{\partial L_H}{\partial \boldsymbol{u}_l}$ can be computed as

$$\frac{\partial L_H}{\partial \boldsymbol{u}_l} = \frac{\partial L_H}{\partial \boldsymbol{u}_{l+1}}\frac{\partial \boldsymbol{u}_{l+1}}{\partial \boldsymbol{o}_l}\frac{\partial \boldsymbol{o}_l}{\partial \boldsymbol{z}_l}\frac{\partial \boldsymbol{z}_l}{\partial \boldsymbol{u}_l} + \frac{\partial H(\boldsymbol{u}_l)}{\partial \boldsymbol{u}_l} = \frac{\partial L_H}{\partial \boldsymbol{u}_{l+1}}\boldsymbol{w}_{l+1}\frac{\partial \boldsymbol{o}_l}{\partial \boldsymbol{z}_l}\frac{1}{v_l^{th}} + \frac{\partial H(\boldsymbol{u}_l)}{\partial \boldsymbol{u}_l} \tag{8}$$

where $\frac{\partial L_H}{\partial \boldsymbol{u}_{l+1}}$ is the gradient backpropagated from the $(l+1)^{th}$ layer, that is iteratively computed from the last layer $L$ (see Eqs. 6 and 9). Note that for any hidden layer $l$, there are two gradients that contribute to the Hoyer loss with respect to the potential $\boldsymbol{u}_l$; one is from the subsequent layer $(l+1)$ and the other is directly from its Hoyer regularizer. Similarly, $\frac{\partial L_{CE}}{\partial \boldsymbol{o}_l}$ is computed iteratively, starting from the penultimate layer $(L-1)$ defined in Eq. 6, as follows.

$$\frac{\partial L_{CE}}{\partial \boldsymbol{o}_l} = \frac{\partial L_{CE}}{\partial \boldsymbol{o}_{l+1}}\frac{\partial \boldsymbol{o}_{l+1}}{\partial \boldsymbol{z}_{l+1}}\frac{\partial \boldsymbol{z}_{l+1}}{\partial \boldsymbol{u}_{l+1}}\frac{\partial \boldsymbol{u}_{l+1}}{\partial \boldsymbol{o}_l} = \frac{\partial L_{CE}}{\partial \boldsymbol{o}_{l+1}}\frac{\partial \boldsymbol{o}_{l+1}}{\partial \boldsymbol{z}_{l+1}}\frac{1}{v_l^{th}}\boldsymbol{w}_{l+1} \tag{9}$$

All the derivatives in Eq. 8-11 can be computed by Pytorch autograd, except the spike derivative $\frac{\partial \boldsymbol{o}_l}{\partial \boldsymbol{z}_l}$, whose gradient is zero almost everywhere and undefined at $o_l=0$. We extend the existing idea of surrogate gradient (Neftci et al., 2019) to compute this derivative for one-time-step SNNs with Hoyer spike layers, as illustrated in Fig. 1(b) and mathematically defined as follows.

$$\frac{\partial \boldsymbol{o}_l}{\partial \boldsymbol{z}_l} = \begin{cases} scale \times 1 & \text{if } 0 < \boldsymbol{z}_l < 2 \\ 0 & \text{otherwise} \end{cases} \tag{10}$$

where *scale* denotes a hyperparameter that controls the dampening of the gradient. Finally, the threshold update for the hidden layer $l$ is computed as

$$\Delta v_l^{th} = \frac{\partial L_{CE}}{\partial v_l^{th}} + \lambda_H \frac{\partial L_H}{\partial v_l^{th}} = \frac{\partial L_{CE}}{\partial \boldsymbol{o}_l}\frac{\partial \boldsymbol{o}_l}{\partial \boldsymbol{z}_l}\frac{\partial \boldsymbol{z}_l}{\partial v_l^{th}} + \lambda_H \frac{\partial L_H}{\partial v_l^{th}} = \frac{\partial L_{CE}}{\partial \boldsymbol{o}_l}\frac{\partial \boldsymbol{o}_l}{\partial \boldsymbol{z}_l}\frac{-\boldsymbol{u}_l}{(v_l^{th})^2} + \lambda_H \frac{\partial L_H}{\partial \boldsymbol{u}_{l+1}}\frac{\partial \boldsymbol{u}_{l+1}}{\partial v_l^{th}} \tag{11}$$

$$\frac{\partial \boldsymbol{u}_{l+1}}{\partial v_l^{th}} = \frac{\partial \boldsymbol{u}_{l+1}}{\partial \boldsymbol{o}_l} \cdot \frac{\partial \boldsymbol{o}_l}{\partial v_l^{th}} = \boldsymbol{w}_{l+1} \cdot \frac{\partial \boldsymbol{o}_l}{\partial \boldsymbol{z}_l} \cdot \frac{-\boldsymbol{u}_l}{(v_l^{th})^2} \tag{12}$$

Note that we use this $v_{th}^l$, which is updated in each iteration, to estimate the threshold value in our spiking model using Eq. 3.

### 3.3 NETWORK STRUCTURE

We propose a series of network architectural modifications of existing SNNs (Datta & Beerel, 2022; Chowdhury et al., 2021; Rathi et al., 2020a) for our one-time-step models. As shown in Fig. 2(a), for the VGG variant, we use the max pooling layer immediately after the convolutional layer that is common in many BNN architectures (Rastegari et al., 2016), and introduce the BN layer after max pooling. Similar to recently developed multi-time-step SNN models (Zheng et al., 2021; Li et al., 2021b; Deng et al., 2022; Meng et al., 2022), we observe that BN helps increase the test accuracy with one time step. In contrast, for the ResNet variants, inspired by (Liu et al., 2018), we observe models with shortcuts that bypass every block can also further improve the performance of the SNN. We also observe that the sequence of BN layer, Hoyer spike layer, and convolution layer outperforms the original bottleneck in ResNet. More details are shown in Fig. 2(b).

### 3.4 POSSIBLE TRAINING STRATEGIES

Based on existing SNN literature, we hypothesize a couple of training strategies that can used to train one-time-step SNNs, other than our proposed approach.

Figure 2: Spiking network architectures corresponding to (a) VGG and (b) ResNet based models.

**Pre-trained DNN, followed by SNN fine-tuning.** Similar to the hybrid training proposed in (Rathi et al., 2020b), we pre-train a non-spiking DNN model, and copy its weights to the SNN model. Initialized with these weights, we train a one-time-step SNN with normal cross-entropy loss.

**Iteratively convert ReLU neurons to spiking neurons.** First, we train a DNN model which uses the ReLU function with threshold as the activation function, then we iteratively reduce the number of the ReLU neurons whose output activation values are multi-bit. Specifically, we first force the neurons with values in the top $N$ percentile to spike (set the output be 1), and those with bottom $N$ percentile percent to die (set the output be 0), and gradually increase the $N$ until there is a significant drop of accuracy or all neuron outputs are either 1 or 0.

**Proposed training from scratch.** With our proposed Hoyer spike layer and Hoyer loss, we train a SNN model from scratch. Our results with these training strategies are shown in Table 1, which indicates that it is difficult for training strategies that involve pre-training and fine-tuning to approach the

Table 1: Accuracies from different strategies to train one-step SNNs on CIFAR10

| Training Strategies | Pretrained DNN(%) | Acc. (%) | Spiking activity (%) |
|---|---|---|---|
| Pre-trained+fine-tuning | 93.15 | 91.39 | 23.56 |
| Iterative training (N=10) | 93.25 | 92.68 | 10.22 |
| Iterative Training (N=20) | 92.68 | 92.24 | 9.54 |
| Proposed Training | - | **93.13** | 22.57 |

accuracy of non-spiking models with one time step. One possible reason for this might be the difference in the distribution of the pre-activation values between the DNN and SNN models (Datta & Beerel, 2022). It is also intuitive to obtain a one-time-step SNN model by iteratively reducing the proportion of the ReLU neurons from a pretrained full-precision DNN model. However, our results indicate that this method also fails to generate enough spikes at one time step required to yield close to SOTA accuracy. Finally, with our network structure modifications to existing SNN works, our Hoyer spike layer and our Hoyer regularizer, we can train a one-time-step SNN model with SOTA accuracy from scratch.

## 4 EXPERIMENTAL RESULTS

**Datasets & Models**: Similar to existing SNN works (Rathi et al., 2020b;a), we perform object recognition experiments on CIFAR10 (Krizhevsky et al., 2009) and ImageNet (Deng et al., 2009) dataset using VGG16 (Simonyan & Zisserman, 2014) and several variants of ResNet (He et al., 2016) architectures. For object detection, we use the MMDetection framework (Chen et al., 2019) with PASCAL VOC2007 and VOC2014 (Everingham et al., 2010) as training dataset, and benchmark our SNN models and the baselines on the VOC2007 test dataset. We use the Faster R-CNN (Ren et al., 2015) and RetinaNet (Lin et al., 2017) framework, and substitute the original backbone with our SNN models that are pretrained on ImageNet1K.

**Object Recognition Results**: For training the recognition models, we use the Adam (Kingma & Ba, 2014) optimizer for VGG16, and use SGD optimizer for ResNet models. As shown in Table 2, we obtain the SOTA accuracy of $93.44\%$ on CIFAR10 with VGG16 with only one time step; the accuracy of our ResNet-based SNN models on ImageNet also surpasses the existing works. On ImageNet, we obtain a $68.00\%$ top-1 accuracy with VGG16 which is only $\sim 2\%$ lower compared to the non-spiking counterpart. All our SNN models yield a spiking activity of $\sim 25\%$ or lower on both CIFAR10 and ImageNet, which is significantly lower compared to the existing multi-time-step SNN models as shown in Fig. 3.

Table 2: Comparison of the test accuracy of our one-time-step SNN models with the non-spiking DNN models for object recognition. Model* indicates that we remove the first max pooling layer.

| Network | dataset | DNN Top 1(%) | SNN Top 1 (%) | SNN Top 5 (%) | Spiking activity (%) |
|---|---|---|---|---|---|
| VGG16 | CIFAR10 | 94.10 | 93.44 | 97.88 | 21.87 |
| ResNet18 | CIFAR10 | 93.34 | 91.48 | 97.34 | 25.83 |
| ResNet18* | CIFAR10 | 94.28 | 93.67 | 97.98 | 16.12 |
| ResNet20 | CIFAR10 | 93.18 | 92.38 | 97.63 | 23.69 |
| ResNet34* | CIFAR10 | 94.68 | 93.47 | 97.86 | 16.04 |
| ResNet50* | CIFAR10 | 94.90 | 93.00 | 97.86 | 17.79 |
| VGG16 | ImageNet | 70.08 | 68.00 | 78.75 | 24.48 |
| ResNet50 | ImageNet | 73.12 | 66.32 | 77.06 | 23.89 |

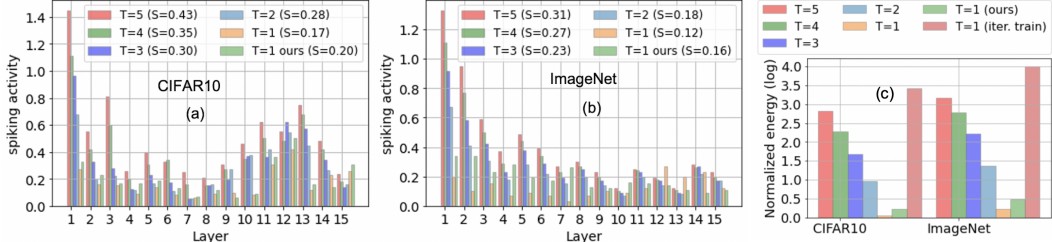

Figure 3: Layerwise spiking activities for a VGG16 across time steps ranging from 5 to 1 (average spiking activity denoted as $S$ in parenthesis) representing existing low-latency SNNs including our work on (a) CIFAR10, (b) ImageNet, (c) Comparison of the total energy consumption between SNNs with different time steps and non-spiking DNNs.

**Object Detection Results**: For object detection on VOC2007, we compare the performance obtained by our spiking models with non-spiking DNNs and BNNs in Table 3. For two-stage architectures, such as Faster R-CNN, the mAP of our one-time-step SNN models surpass the existing BNNs by $>0.6\%$[1]. For one-stage architectures, such as RetinaNet (chosen because of its SOTA performance), our one-time-step SNN models with a ResNet50 backbone

Table 3: Comparison of our one-time-step SNN models with non-spiking DNN, BNN, and multi-step SNN counterparts on VOC2007 test dataset.

| Framework | Backbone | mAP(%) |
|---|---|---|
| Faster R-CNN | Original ResNet50 | 79.5 |
| Faster R-CNN | Bi-Real (Liu et al., 2018) | 65.7 |
| Faster R-CNN | ReActNet (Liu et al., 2020) | 73.1 |
| Faster R-CNN | Our spiking ResNet50 | 73.7 |
| Retinanet | Original ResNet50 | 77.3 |
| Retinanet | SNN ResNet50 (ours) | 70.5 |
| YOLO | SNN DarkNet (Kim et al., 2019) | 53.01 |
| SSD | BNN VGG16 (Wang et al., 2020c) | 66.0 |

yields a mAP of $70.5\%$ (highest among existing BNN, SNN, AddNNs). Note that our spiking VGG and ResNet-based backbones lead to a significant drop in mAP with the YOLO framework that is more compatible with the DarkNet backbone (even existing DarkNet-based SNNs lead to very low mAP with YOLO as shown in Table 3). However, our models suffer $5.8-6.8\%$ drop in mAP compared to the non-spiking DNNs which may be due to the significant sparsity and loss in precision.

**Accuracy Comparison**: We compare our results with various SOTA ultra low-latency SNNs for image recognition tasks in Table 4. Our one-time-step SNNs yield comparable or better test accuracy compared to all the existing works for both VGG and ResNet architectures, with significantly lower inference latency. The only exception for the latency reduction is the one-time-step SNN proposed in (Chowdhury et al., 2021), however, it increases the training time significantly as illustrated later in Fig. 3. Other works that have training complexity similar or worse than ours, such as (Datta & Beerel, 2022) yields $1.78\%$ lower accuracy with a $2\times$ more number of time steps. Across both CIFAR10 and ImageNet, our proposed training framework demonstrates $2-32\times$ improvement in inference latency with similar or worse training complexity compared to other works while yielding better test accuracy. Table 4 also demonstrates that the DNN-SNN conversion approaches require more time steps compared to our approach at worse test accuracies.

---

[1]We were unable to find existing SNN works for two-stage object detection architectures.

Table 4: Comparison of our one-time-step SNN models to existing low-latency counterparts. SGD and hybrid denote surrogate gradient descent and pre-trained DNN followed by SNN fine-tuning respectively. (qC, dL) denotes an architecture with q convolutional and d linear layers.

| Reference | Training | Architecture | Acc. (%) | Time steps |
|---|---|---|---|---|
| Dataset : CIFAR10 | | | | |
| (Deng et al., 2021) | DNN-SNN conversion | VGG16 | 92.29 | 16 |
| (Wu et al., 2019) | SGD | 5C, 2L | 90.53 | 12 |
| (Kundu et al., 2021) | Hybrid | VGG16 | 92.74 | 10 |
| (Wu et al., 2021) | Tandem Learning | 5C, 2L | 90.98 | 8 |
| (Bu et al., 2022a) | DNN-SNN coonversion | VGG16 | 90.96 | 8 |
| (Zhang & Li, 2020) | SGD | 5C, 2L | 91.41 | 5 |
| (Rathi et al., 2020a) | Hybrid | VGG16 | 92.70 | 5 |
| (Zheng et al., 2021) | STBP-tdBN | ResNet19 | 93.16 | 6 |
| (Datta & Beerel, 2022) | Hybrid | VGG16 | 91.79 | 2 |
| (Bu et al., 2022b) | DNN-SNN conversion | VGG16 | 91.18 | 2 |
| (Fang et al., 2020) | SGD | 5C, 2L | **93.50** | 8 |
| (Chowdhury et al., 2021) | Hybrid | VGG16 | 93.05 | 1 |
| (Chowdhury et al., 2021) | Hybrid | ResNet20 | 91.10 | 1 |
| **This work** | **Adam+Hoyer Reg.** | **VGG16** | **93.44** | **1** |
| Dataset : ImageNet | | | | |
| (Li et al., 2021c) | DNN-SNN conversion | VGG16 | 63.64 | 32 |
| (Bu et al., 2022b) | DNN-SNN conversion | ResNet34 | 59.35 | 16 |
| (Wu et al., 2021) | Tandem Learning | AlexNet | 50.22 | 12 |
| (Rathi et al., 2020a) | Hybrid | VGG16 | 69.00 | 5 |
| (Fang et al., 2021) | SGD | ResNet34 | 67.04 | 4 |
| (Fang et al., 2021) | SGD | ResNet152 | **69.26** | 4 |
| (Rathi et al., 2020a) | Hybrid | VGG16 | 69.00 | 5 |
| (Zheng et al., 2021) | STBP-tdBN | ResNet34 | 67.05 | 6 |
| (Chowdhury et al., 2021) | Hybrid | VGG16 | 67.71 | 1 |
| **This work** | **Adam+Hoyer Reg.** | **VGG16** | **68.00** | **1** |

**Inference Efficiency**: We compare the energy-efficiency of our one-time-step SNNs with non-spiking DNNs and existing multi-time-step SNNs in Fig. 3. The compute-efficiency of SNNs stems from two factors:- 1) sparsity, that reduces the number of floating point operations in convolutional and linear layers compared to non-spiking DNNs according to $SNN_l^{flops} = S_l \times DNN_l^{flops}$ (Chowdhury et al., 2021), where $S_l$ denotes the average number of spikes per neuron per inference over all timesteps in layer $l$. 2) Use of only AC (0.9pJ) operations that consume $5.1\times$ lower compared to each MAC (4.6pJ) operation in 45nm CMOS technology (Horowitz, 2014) for floating-point (FP) representation. Note that the binary activations can replace the FP multiplications with logical operations, i.e., conditional assignment to 0 with a bank of AND gates. These replacements can be realized using existing hardware (eg. standard GPUs) depending on the compiler and the details of their data paths. Building a custom accelerator that can efficiently implement these reduced operations is also possible (Wang et al., 2020a; Frenkel et al., 2019; Lee & Li, 2020). In fact, in neuromorphic accelerators such as Loihi (Davies et al., 2018), FP multiplications are typically avoided using message passing between processors that model multiple neurons.

The total compute energy (CE) of a multi-time-step SNN ($SNN_{CE}$) can be estimated as

$$SNN_{CE} = DNN_1^{flops} * 4.6 + DNN_1^{com} * 0.4 + \sum_{l=2}^{L} S_l * DNN_l^{flops} * 0.9 + DNN_l^{com} * 0.7$$

(13)

because the direct encoded SNN receives analog input in the first layer ($l=1$) without any sparsity (Chowdhury et al., 2021; Datta & Beerel, 2022; Rathi et al., 2020a). Note that $DNN_l^{com}$ denotes the total number of comparison operations in the layer $l$ with each operation consuming 0.4pJ energy. The CE of the non-spiking DNN ($DNN_{CE}$) is estimated as $DNN_{CE} = \sum_{l=1}^{L} DNN_l^{flops} * 4.6$, where we ignore the energy consumed by the ReLU operation since that includes only checking the sign bit of the input.

We compare the layer-wise spiking activities $S_l$ for time steps ranging from 5 to 1 in Fig. 3(a-b) that represent existing low-latency SNN works, including our work. Note, the spike rates decrease significantly with time step reduction from 5 to 1, leading to considerably lower FLOPs in our one-

time-step SNNs. These lower FLOPs, coupled with the $5.1\times$ reduction for AC operations leads to a $22.9\times$ and $32.1\times$ reduction in energy on CIFAR10 and ImageNet respectively with VGG16. Though we focus on compute energies for our comparison, multi-time-step SNNs also incur a large number of memory accesses as the membrane potentials and weights need to be fetched from and read to the on-/off-chip memory for each time step. Our one-time-step models can avoid these repetitive read/write operations as it does involve any *state* and lead to a $\sim T\times$ reduction in the number of memory accesses compared to a $T$-time-step SNN model. Considering this memory cost and the overhead of sparsity (Yin et al., 2022), as shown in Fig. 3(c), our one-time-step SNNs lead to a $2.08-14.74\times$ and $22.5-31.4\times$ reduction of the total energy compared to multi-time-step SNNs and non-spiking DNNs respectively on a systolic array accelerator.

**Training & Inference Time Requirements**: Because SOTA SNNs require iteration over multiple time steps and storage of the membrane potentials for each neuron, their training and inference time can be substantially higher than their DNN counterparts. However, reducing their latency to 1 time step can bridge this gap significantly, as shown in Figure 4. On average, our low-latency, one-time-step SNNs represent a $2.38\times$ and $2.33\times$ reduction in training and inference time per epoch respectively, compared to the multi-time-step training approaches (Datta & Beerel, 2022; Rathi et al.,

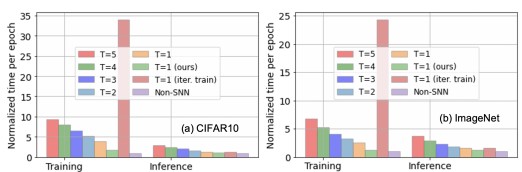

Figure 4: Normalized training and inference time per epoch with iso-batch (256) and hardware (RTX 3090 with 24 GB memory) conditions for (a) CIFAR10 and (b) ImageNet with VGG16.

2020a) with iso-batch and hardware conditions. Compared to the existing one-time-step SNNs (Chowdhury et al., 2021), we yield a $19\times$ and $1.25\times$ reduction in training and inference time. Such significant savings in training time, which translates to power savings in big data centers, can potentially reduce AI's environmental impact.

**Ablation Studies**: We conduct ablation studies to analyze the contribution of each technique in our proposed approach. For fairness, we train all the ablated models on CIFAR10 dataset for 400 epochs, and use Adam as the optimizer, with 0.0001 as the initial learning rate. Our results are shown in Table 5, where the model without Hoyer spike layer indicates that we set the threshold as $v_l^{th}$ similar to existing works (Datta & Beerel, 2022; Rathi et al., 2020a) rather than our proposed Hoyer extremum. With VGG16, our op-

Table 5: Ablation study of the different methods in our proposed training framework on CIFAR10.

| Arch. | Network Structure | Hoyer Reg. | Hoyer Spike | Acc. (%) | Spiking Activity (%) |
|---|---|---|---|---|---|
| VGG16 | $\times$ | $\times$ | $\times$ | 88.42 | 15.62 |
| VGG16 | $\checkmark$ | $\times$ | $\times$ | 90.33 | 20.43 |
| VGG16 | $\checkmark$ | $\checkmark$ | $\times$ | 90.45 | 20.48 |
| VGG16 | $\checkmark$ | $\times$ | $\checkmark$ | 92.90 | 21.70 |
| VGG16 | $\checkmark$ | $\checkmark$ | $\checkmark$ | **93.13** | 22.57 |
| ResNet18 | $\times$ | $\times$ | $\times$ | 87.41 | 22.78 |
| ResNet18 | $\checkmark$ | $\times$ | $\times$ | 91.08 | 27.62 |
| ResNet18 | $\checkmark$ | $\checkmark$ | $\times$ | 90.95 | 20.50 |
| ResNet18 | $\checkmark$ | $\times$ | $\checkmark$ | 91.17 | 25.87 |
| ResNet18 | $\checkmark$ | $\checkmark$ | $\checkmark$ | **91.48** | 25.83 |

timal network modifications lead to a $1.9\%$ increase in accuracy. Furthermore, adding only the Hoyer regularizer leads to negligible accuracy and spiking activity improvements. This might be because the regularizer alone may not be able to sufficiently down-scale the threshold for optimal convergence with one time step. However, with our Hoyer spike layer, the accuracy improves by $2.68\%$ to $93.13\%$ while also yielding a $2.09\%$ increase in spiking activity. We observe a similar trend for our network modifications and Hoyer spike layer with ResNet18. However, Hoyer regularizer substantially reduces the spiking activity from $27.62\%$ to $20.50\%$, while also negligibly reducing the accuracy. Note that the Hoyer regularizer alone contributes to $0.20\%$ increase in test accuracy on average. In summary, while our network modifications significantly increase the test accuracy compared to the SOTA SNN training with one time step, the combination of our Hoyer regularizer and Hoyer spike layer yield the SOTA SNN performance.

**Effect on Quantization**: In order to further improve the compute-efficiency of our one-time-step SNNs, we perform quantization-aware training of the weights in our models to $2-6$ bits.

This transforms the full-precision ACs to $2-6$ bit ACs, thereby leading to a $4.8-13.8$ reduction in compute energy as obtained from FPGA simulations on the Kintex7 platform using custom RTL specifications. Note that we only quantize the convolutional layers, as quantizing the linear layers lead to a noticeable drop in accuracy. From the results shown in Table 6, when quantized to 6 bits, our one-time-step VGG-based SNN incur a negligible accuracy drop of only $0.02\%$. Even with 2-bit quantization, our model can yield an accuracy of $92.34\%$ with any special modification, while still yielding a spiking activity of $\sim 22\%$.

Table 6: Accuracies of weight quantized one-time-step SNN models based on VGG16 on CIFAR10 where FP is 32-bit floating point.

| Bits | Acc. (%) | Spiking Activity (%) | CE (mJ) |
|------|----------|----------------------|---------|
| FP   | 93.13    | 22.57                | 297.42  |
| 6    | 93.11    | 22.46                | 61.9    |
| 4    | 92.84    | 21.39                | 39.4    |
| 2    | 92.34    | 22.68                | 21.6    |

**Comparison with AddNNs & BNNs**: We compare the accuracy and CE of our one-time-step SNN models with recently proposed AddNN models (Chen et al., 2020) that also removes multiplications for increased energy-efficiency in Table 7. With the VGG16 architecture, on CIFAR10, we obtain $0.6\%$ lower accuracy, while on ImageNet, we obtain $1.0\%$ higher accuracy. Moreover, unlike SNNs, AddNNs do not involve any sparsity, and hence, consume $\sim 5.5\times$ more energy compared to our SNN models on average across both CIFAR10 and ImageNet (see Table 7). We also compare our SNN models with SOTA BNNs in Table 7 that replaces the costly MAC operations with cheaper pop-count counterparts, thanks to the binary weights and activations. Both our

Table 7: Comparison of our one-time-step SNN models to AddNNs and BNNs that also incur AC-only operations for improved energy-efficiency, where CE is compute energy

| Reference | Dataset | Acc.(%) | CE (J) |
|-----------|---------|---------|--------|
| BNNs | | | |
| Sakr et al. (2018) | CIFAR10 | 89.6 | 0.022 |
| Wang et al. (2020b) | CIFAR10 | 90.2 | 0.019 |
| Wang et al. (2020b) | ImageNet | 59.7 | 3.6 |
| Diffenderfer & Kailkhura (2021) | CIFAR10 | 91.9 | 0.073 |
| AddNNs | | | |
| Chen et al. (2020) (FP weights) | CIFAR10 | 93.72 | 1.62 |
| Chen et al. (2020) (2-bit weights) | CIFAR10 | 92.08 | 0.12 |
| Chen et al. (2020) (FP weights) | ImageNet | 67.0 | 77.8 |
| Li et al. (2021a) (FP weights) | CIFAR10 | 91.56 | 1.62 |
| Our SNNs | | | |
| This work (FP weights) | CIFAR10 | 93.44 | 0.297 |
| This work (2-bit weights) | CIFAR10 | 92.34 | 0.021 |
| This work (FP weights) | ImageNet | 68.00 | 14.28 |

full-precision and 2-bit quantized one-time-step SNN models yield accuracies higher than BNNs at iso-architectures on both CIFAR10 and ImageNet. Additionally, our 2-bit quantized SNN models also consume $3.4\times$ lower energy compared to the bi-polar networks (see (Diffenderfer & Kailkhura, 2021) in Table 7) due to the improved trade-off between the low spiking activity ($\sim 22\%$ as shown in Table 7) provided by our one-time-step SNN models, and less energy due to XOR operations compared to quantized ACs. On the other hand, our one-time-step SNNs consume similar energy compared to unipolar BNNs (see (Sakr et al., 2018; Wang et al., 2020b) in Table 7) while yielding $3.2\%$ higher accuracy on CIFAR10 at iso-architecture. The energy consumption is similar because the $\sim 20\%$ advantage of the pop-count operations is mitigated by the $\sim 22\%$ higher spiking activity of the unipolar BNNs compared to our one-time-step SNNs.

## 5 DISCUSSIONS & FUTURE IMPACT

Existing SNN training works choose ANN-SNN conversion methods to yield high accuracy or SNN fine-tuning to yield low latency or a hybrid of both for a balanced accuracy-latency trade-off. However, none of the existing works can discard the temporal dimension completely, which can enable the deployment of SNN models in multiple real-time applications, without significantly increasing the training cost. This paper presents a SNN training framework from scratch involving a novel combination of a Hoyer regularizer and Hoyer spike layer for one time step. Our SNN models incur similar training time as non-spiking DNN models and achieve SOTA accuracy, outperforming the existing SNN, BNN, and AddNN models. However, our work can also enable cheap and real-time computer vision systems that might be susceptible to adversarial attacks. Preventing the application of this technology from abusive usage is an important and interesting area of future work.

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

# A APPENDIX

## A.1 PROOF OF THRESHOLD DOWNSACLING WITH HOYER EXTREMUM

In order to prove that our Hoyer extremum is always less than or equal to $v^{th}$, we first prove the Hoyer extremum of $z_l^{clip}$ is less than or equal to 1. Let us use $c_l$ to represent $z_l^{clip}$, so $\forall j, 0 \leq c_l^j \leq 1$

$$Ext(c_l) = \frac{\|c_l\|_2^2}{\|c_l\|_1} = \frac{\sum_j (c_l^j)^2}{\sum_j c_l^j} \leq \frac{\sum_j (c_l^j \cdot max(c_l^j))}{\sum_j c_l^j} \leq max(c_l^j)) \leq 1 \qquad (14)$$

So the Hoyer extremum of $z_l^{clip}$ is always less than or equal one, and our Hoyer extremum of every layer $l$ which is the product of $v_l^{th}$ and $Ext(z_l^{clip})$ is always less than or equal $v_l^{th}$.

## A.2 EXPERIMENTAL SETUP

For training VGG16 models, we using Adam optimizer with initial learning rate of 0.0001, weight decay of 0.0001, dropout of 0.1 and batch size of 128 in CIFAR10 for 600 epochs, and Adam optimizer with weight decay of $5e^{-6}$ and with batch size 64 in ImageNet for 180 epochs. For training ResNet models, we using SGD optimizer with initial learning rate of 0.1, weight decay of 0.0001 and batch size of 128 in CIFAR10 for 400 epochs, and Adam optimizer with weight decay of $5e^{-6}$ and with batch size 64 in ImageNet for 120 epochs. We divide the learning rate by 5 at $60\%$, $80\%$, and $90\%$ of the total number of epochs.

When calculating the Hoyer extremum we implement two versions, one that calculates the Hoyer extremum for the whole batch, while another that calculates it channel-wise. Our experiments show that using the channel-wise version can bring $0.1-0.3\%$ increase in accuracy. All the experimental results reported in this paper use this channel-wise version.

For Faster R-CNN, we use SGD optimizer with initial learning rate of 0.01 for 50 epochs, and divide the learning rate by 10 after 25 and 40 epochs each. For Retinanet, we use SGD optimizer with initial learning rate of 0.001 with the same learning rate scheduler as Faster R-CNN.

## A.3 EXTENSION TO MULTIPLE TIME-STEPS

We extend our proposed approach to multi-time-step SNN models. As show in Table 8, as time step increases from 1 to 4, the accuracy of the model also increases from 93.44% to 94.14%, which validates the effectiveness of our method. However, this accuracy increase comes at the cost of a significant increase in spiking activity (see Table 8), thereby increasing the compute energy and the temporal "overhead" increases, thereby increasing the memory cost due to the repetitive access of the membrane potential and weights across the different time steps.

## A.4 EXTENSION TO DYNAMIC VISION SENSOR (DVS) DATASETS

The inherent temporal dynamics in SNNs may be better leveraged in DVS or event-based tasks (Deng et al., 2022; Li et al., 2022; Kim & Panda, 2021; Kim et al., 2022a) compared to standard static vision tasks that are studied in this work. Hence, we have evaluated our framework on the DVS-CIFAR10 dataset, which provides each label with only 0.9k training samples, and is considered

Table 8: Test accuracy obtained by our approach with multiple time steps on CIFAR10.

| Architecture | Time steps | Acc. (%) | Spiking activity (%) |
|---|---|---|---|
| VGG16 | 1 | 93.44 | 21.87 |
| VGG16 | 2 | 93.71 | 44.06 |
| VGG16 | 4 | 94.14 | 74.88 |
| VGG16 | 6 | 94.04 | 101.22 |
| ResNet18 | 1 | 91.48 | 25.83 |
| RseNet18 | 2 | 91.93 | 33.24 |

Table 9: Comparison of our one- and multi-time-step SNN models to existing SNN models on DVS-CIFAR10 dataset.

| Reference | Training | Architecture | Acc. (%) | Time steps |
|---|---|---|---|---|
| Deng et al. (2022) | TET | VGGSNN | 83.17 | 10 |
| Deng et al. (2022) | TET | VGGSNN | 75.20 | 4 |
| Deng et al. (2022) | TET | VGGSNN | 68.12 | 1 |
| Li et al. (2022) | tdBN+NDA | VGG11 | 81.7 | 10 |
| Kim & Panda (2021) | SALT+Switchec BN | VGG16 | 67.1 | 20 |
| This work | Hoyer reg. | VGGSNN | **83.68** | 10 |
| This work | Hoyer reg. | VGGSNN | **76.17** | 4 |
| This work | Hoyer reg. | VGGSNN | **69.80** | 1 |

the most challenging event-based dataset (Deng et al., 2022). As illustrated in Table 9, we surpass the test accuracy of existing works (Li et al., 2022; Kim & Panda, 2021) by 1.30% on average at iso-time-step and architecture. Note that the architecture VGGSNN employed in our work and (Deng et al., 2022) is based on VGG11 with two fully connected layers removed as (Deng et al., 2022) found that additional fully connected layers were unnecessary for neuromorphic datasets. In fact, our accuracy gain is more significant at low time steps, thereby implying the portability of our approach to DVS tasks. Note that similar to static datasets, a large number of time steps increase the temporal overhead in SNNs, resulting in a large memory footprint and spiking activity.

## A.5 FURTHER INSIGHTS ON HOYER REGULARIZED TRAINING

Since existing works (Panda et al., 2020) use surrogate gradients (and not real gradients) to update the thresholds with appropriate initializations, it is difficult to estimate the optimal value of the IF thresholds. On the other hand, our Hoyer extremums dynamically change with the activation maps particularly during the early stages of training (coupled with the distribution shift enabled by Hoyer regularized training), which enables our Hoyer extremum-based scaled thresholds to be closer to optimal. In fact, as shown from our ablation studies in Table 5, our Hoyer extremum-based spike layer is more effective than the Hoyer regularizer which further justifies the importance of the combination of the Hoyer extremum with the trainable threshold. Additionally, we use the clip function of the membrane potential before computing the Hoyer extremum. This is done to get rid of a few outlier values in the activation map that may otherwise unnecessarily increase the value of the Hoyer extremum, i.e., threshold value, thereby reducing the accuracy, without any noticeable increase in energy efficiency. In fact, the test accuracy with VGG16 on CIFAR10 drops by more than 1.4% (from 93.13% obtained by our training framework) to 91.7% without the clip function.

## A.6 TUNING SPIKING ACTIVITY WITH HOYER REGULARIZER $\lambda_H$

We conduct experiments with different coefficients of Hoyer regularizer $\lambda_H$ to demostrate its impact on the trade-off between accuracy and spikinf activity. As shown in Table 10, we can clearly see that a larger Hoyer regularizer alone can decrease the spike activity rate, while a smaller Hoyer regularizer will increase the same. In fact, the spiking activity can be precisely tuned using $\lambda_H$ to yield a range of accuracies. Interestingly, Hoyer-regularized training on ResNet18 yields a wider range of spiking activities and a narrower range of accuracies compared to VGG16. This might be because each architecture can have different optimization headroom.

Table 10: Test accuracy obtained with different coefficients of Hoyer regularizer on CIFAR10.

| Architecture | $\lambda_H$ | Acc. (%) | Spiking activity (%) |
|---|---|---|---|
| VGG16 | 1e-7 | 89.73 | 19.62 |
| VGG16 | 1e-8 | 90.33 | 20.43 |
| VGG16 (with Hoyer spike layer) | 1e-7 | 92.93 | 21.61 |
| VGG16 (with Hoyer spike layer) | 1e-8 | 93.13 | 22.57 |
| VGG16 (with Hoyer spike layer) | 1e-9 | 92.95 | 22.15 |
| ResNet18 | 1e-7 | 90.84 | 13.05 |
| ResNet18 | 1e-8 | 90.95 | 20.50 |
| ResNet18 | 1e-9 | 91.05 | 23.54 |
| ResNet18 (with Hoyer spike layer) | 1e-8 | 91.48 | 25.83 |
| ResNet18 (with Hoyer spike layer) | 0 | 91.17 | 25.87 |

The Hoyer spike layer, when used with the Hoyer regularizer (with the optimal value of the co-efficient that yields the best test accuracy), increase the spiking activity for both VGG16 and ResNet18. Please check the 2.08% (from 20.48% to 22.57%) increase in spiking activity for VGG16 and 5.33% (20.50% to 25.83%) for ResNet18. This is because the Hoyer spike layer downscales the threshold value, enabling more neurons to spike.

Note that the Hoyer spike layer, when used without the Hoyer regularizer, may be unable to tune the trade-off between spiking activity and accuracy. This is because we do not have any explicit regularizer co-efficient, and the Hoyer extremum may not always lower the threshold value because it is computed based on the SGL-based trainable threshold which, without Hoyer regularizer, may be updated randomly (i.e., not in a systematic manner that may encourage sparsity). This is the reason we believe we do not observe any definitive trend for the trade-off between accuracy and spiking activity in this case. Note that all the results in Table 9 are reported as the mean from five runs with distinct seeds.

## A.7 TRAINING ALGORITHM

Our proposed training framework that can yield accurate and sparse one-time-step SNN models is illustrated below.

---

**Algorithm 1:** Detailed Algorithm for training our one-time-step SNN model.

---

**Input**: runEpochs, numBatches, numLayers, initial weights $\mathbf{W}$, initial thresholds $\boldsymbol{v}^{th}$, Training data $\{(\boldsymbol{x}^{(i)}, \boldsymbol{y}^{(i)})\}_{i=1}^{N}$, Hoyer regularizer coefficient $\lambda_H$.
**Data:** Hoyer extremum $Ext = 0$, Layer index $l = 0$

**for** $i \leftarrow 0$ **to** *runEpochs* **do**
    **for** $j \leftarrow 0$ **to** *numBatches* **do**
        $output \leftarrow \boldsymbol{x}$
        **for** $l \leftarrow 0$ **to** *numLayers* **do**
            **if** *layer$_l$ is Hoyer Spike layer* **then**
                $\boldsymbol{u}_l \leftarrow output$
                $\mathcal{L}_H \leftarrow \mathcal{L}_H + H(\boldsymbol{u}_l)$
                $\boldsymbol{z}_l \leftarrow \boldsymbol{u}_l / v_l^{th};$               `// Divide input by threshold`
                $Ext \leftarrow \text{computeExponentialMovingAverage}(Ext, Ext(clamp(\boldsymbol{z}_l, min = 0, max = 1)))$
                $\boldsymbol{o}_l[\boldsymbol{z}_l \geq Ext] \leftarrow 1, \boldsymbol{o}_l[\boldsymbol{z}_l < Ext] \leftarrow 0;$     `// Output spiking activation map`
                $output \leftarrow \boldsymbol{o}_l$
            **end**
            **else**
                $output = \text{layer}_l(output)$
            **end**
        **end**
        $\mathcal{L} = \mathcal{L}_{CE} + \lambda_H * \mathcal{L}_H$
        $\frac{\partial \mathcal{L}}{\partial \mathbf{W}} = \text{computeGradients}(\mathbf{W}, \mathcal{L})$
        $\frac{\partial \mathcal{L}}{\partial \boldsymbol{v}^{th}} = \text{computeGradients}(\boldsymbol{v}^{th}, \mathcal{L})$
        $\text{updateWeightsAndThresholds}(\frac{\partial \mathcal{L}}{\partial \mathbf{W}}, \frac{\partial \mathcal{L}}{\partial \boldsymbol{v}^{th}})$
    **end**
    **for** $l \leftarrow 0$ *numLayers* **do**
        **if** *layer$_l$ is Hoyer Spike layer* **then**
            $\boldsymbol{u}_l \leftarrow output$
            $\boldsymbol{z}_l \leftarrow \boldsymbol{u}_l / v_l^{th}$
            $\boldsymbol{o}_l[\boldsymbol{z}_l \geq Ext] \leftarrow 1, \boldsymbol{o}_l[\boldsymbol{z}_l < Ext] \leftarrow 0;$
            $output \leftarrow \boldsymbol{o}_l$
         **end**
        **else**
            $output = \text{layer}_l(output)$
        **end**
    **end**
**end**

---

