# OpenReview forum: "HOYER REGULARIZER IS ALL YOU NEED FOR EXTREMELY SPARSE SPIKING NEURAL NETWORKS"
_ICLR.cc/2023/Conference — Submitted to ICLR 2023_

### Official Review · Reviewer_YzPx · 2022-10-23

**Confidence:** 5
**Correctness:** 2
**Technical Novelty And Significance:** 2
**Empirical Novelty And Significance:** 2
**Recommendation:** 5

**Clarity, Quality, Novelty And Reproducibility:**

Clarity: the paper is well organized and clearly written.

Reproducibility: it cannot be evaluated since no code is provided. Also, no hyperparameters are given in the manuscript.

Quality and Novelty: As for the weaknesses mentioned in the above section, I found that the significance and novelty are marginal.

**Strength And Weaknesses:**

Strength:\
The proposed model was verified on various datasets (CIFAR10, ImageNet, VOC2007) and compared with previous models including DNNs, BNNs, SNNs in a systematic manner.

Weaknesses:

1. SNN comes into its own when encoding time-varying data, so that it should be able to extract spatiotemporal patterns of event stream through time. Therefore, SNN should be defined in a time domain. The authors proposed time-independent SNNs given the use of a single timestep, which are not SNNs as a matter of fact given the key to SNNs, i.e., dynamic models. Therefore, the advantages of the time-independent SNN over DNNs are not sure. In fact, the model used in this work is a DNN with binary activations rather than SNN.

2. The CE comparison between the proposed model and DNN in Eq. 14 is not fair. The proposed model includes additional operations that are not addressed in the CE calculation. They include the division in Eq. 3 and comparison with the threshold in FP in Eq. 4. Note that the comparison with the FP number is much more expansive than the comparison in ReLU, which can simply be done by seeing the sign bit of an input. Moreover, I wonder if FP multiplications are avoided by using binary activations (0/1 or 0.0000…/1.0000…) because this depends on hardware used. I recommend the authors to specify the hardware considered for the proposed model.

3. Where is the multiplier $\lambda_H$ in Eq.6?

**Summary Of The Paper:**

The authors propose an SNN model with one timestep only. The key to the model is the use of the Hoyer extremum as a spiking threshold. The Hoyer regularization is a technique previously used for DNN regularization. The proposed SNN corresponds to a DNN with binary activation functions. The backprop pipeline includes the gradient of the threshold function (nondifferentiable), which is approximated to a simple boxcar function with a sufficient width (0-2) to cover the range of inputs. The performance of the proposed model was evaluated in terms of classification accuracy (on CIFAR 10/ImageNet and VOC2007) and computational efficiency. Given the use of binary activation functions, accuracy degradation was inevitable compared with the DNN counterpart. Instead, the authors place emphasis on computational efficiency that may be enhanced given the use of binary activations that may avoid FP multiplications, this is very arguable though.

**Summary Of The Review:**

The proposed model (SNN with one timestep) corresponds to a DNN with binary activations. Therefore, the model cannot leverage the rich dynamics inherently given to SNNs, and thus the proposed model is not able to process time-varying event stream. Given this reason, I found marginal significance in this work. As well, Novelty is limited given that the Hoyer regularization concept was proposed previously. Regarding technical aspects of the proposed model, the loss in accuracy due to the use of binary activations is obvious, and the computational efficiency estimation is also very arguable.

---

> ### Author Response · Authors · 2022-11-17
> **Response to reviewer YzPx [1/2]**
>
> Thanks for your valuable comments and suggestions to improve the quality of our work. Please see our response below and the revised version of the manuscript.
>
> [Concerns regarding significance and similarity with BNNs]
>
> Please refer to the 'Concerns regarding lack of bio-plausibility/temporal dynamics of one-time-step SNNs' section of the 'Response to all reviewers' comment above for your concerns regarding the advantages of time-independent SNNs. In particular, **our time-independent SNNs offer significantly more sparsity and energy efficiency than DNNs and existing bipolar binary neural networks (BNNs)** as described in the subsection entitled 'Comparison with AddNNs and BNNs' of Section 4.
>
> We would also like to clarify that our approach can process time-varying event streams as well. Upon the request of reviewer hEby, we have now evaluated our framework on the DVS-CIFAR10 dataset, which provides each label with only 0.9k training samples, and is considered the most challenging event-based dataset [6]. **We surpass the test accuracy of existing works [7-9] by 1.07\%** on average at iso-time-step and architecture as shown in Table 9 in Appendix A.4 and below. In fact, **our accuracy gain is more significant at low time steps**, thereby implying the portability of our approach to DVS tasks. Note that similar to static datasets, a large number of time steps in these event-based tasks increase the temporal overhead in SNNs, resulting in a large memory footprint and spiking activity. Hence, a small number of time steps may still be warranted here.
>
> | **Reference** | **Training**         | **Architecture** | **Acc. (%)**  | **Time steps** |
> |-----------|------------------|--------------|-----------|------------|
> | [6]       | TET              | VGGSNN       | 83.17     | 10         |
> | [6]       | TET              | VGGSNN       | 75.20     | 4          |
> | [6]       | TET              | VGGSNN       | 68.12     | 1          |
> | [7]       | tdBN+NDA         | VGG11        | 81.7      | 10         |
> | [8]       | SALT+Switched BN | VGG16        | 67.1      | 20         |
> | This work | Hoyer reg.       | VGGSNN       | **83.68** | 10         |
> | This work | Hoyer reg.       | VGGSNN       | **76.17** | 4          |
> | This work | Hoyer reg.       | VGGSNN       | **69.80** | 1          |
>
> [Unfair comparison of CE]
>
> We agree with the reviewer that our compute energy only includes the accumulate operations in the convolutional and linear layers. However, **the other operations, such as threshold comparison in the IF layers, constitute only a small portion of the total energy**.
>
> The division in Eq. 3 can actually be avoided and the membrane potential can be compared with the threshold instead of 1. We show the division in our IF model for the ease of derivation of the different gradients with respect to our Hoyer loss (see Eq. 3.2). Although the comparison with a floating point number is more expensive than a ReLU operation, **the number of floating point comparisons is significantly less (<0.5\%) than the number of AC operations** in an SNN. A similar comparison holds good for ReLU and MAC operations in a non-spiking DNN as well. In fact, we can quantize the threshold values to 8-bit precisions such that the floating point comparisons are converted to cheaper integer comparisons with <0.1\% loss in accuracy for the CIFAR10 dataset.
>
> Nevertheless, with the incorporation of floating point comparison operations in our energy model, our one-time-step SNNs are still 22.9-32.1$\times$ (24.3-34.2$\times$ was reported in the original paper) more compute-efficient compared to DNNs. Upon the request of reviewer hEby, we have also included the memory cost and the overhead of sparsity in our energy model. Since our one-time-step SNNs avoid the repetitive memory access of the membrane potential and weights across multiple time steps, they show 2.08-14.74$\times$ more energy efficiency compared to multi-time-step SNNs with this improved energy model. We have updated the subsection 'Inference Efficiency' of Section 4.1, Fig 3c, and Table 7 (where we compare the compute energy of our one-time-step SNNs with AddNNs and BNNs) to reflect these changes.

---

> > ### Author Response · Authors · 2022-11-17
> > **References**
> >
> > [1] D. Wang et al. "Always-On, Sub-300-nW, Event-Driven Spiking Neural Network based on Spike-Driven Clock-Generation and Clock- and Power-Gating for an Ultra-LowPower Intelligent Device", IEEE Asian Solid-State Circuits Conference (A-SSCC), 2020
> >
> > [2] C. Frenkel et al, "A 0.086-mm² 12.7-pJ/SOP 64kSynapse 256-Neuron Online-Learning Digital Spiking Neuromorphic Processor in 28-nm CMOS",  IEEE Transactions on Biomedical Circuits and Systems, 2019.
> >
> > [3] Jeong-Jun Lee et al. "Reconfigurable dataflow optimization for spatiotemporal spiking neural computation on systolic array accelerators", IEEE 38th International Conference on Computer Design (ICCD), 2020.
> >
> > [4] Davies et al. "Loihi: A Neuromorphic Manycore Processor with On-Chip Learning", IEEE Micro 2018.

---

> > ### Author Response · Authors · 2022-11-18
> > **Response to reviewer YzPx [2/2]**
> >
> > [Avoiding FP operations using binary activations]
> >
> > Algorithmically, it is clear that the binary activations can be implemented using a single bit, and the **FP multiplications can be replaced with logical operations, i.e., conditional assignment to 0 with a bank of AND gates**. Alternatively, the binary activations can also be captured by zero-gating logic, which also avoids FP multiplications. Whether or not these gains can be realized using existing hardware (eg. standard GPUs) depends on the compiler and the details of their data paths. **Building a custom accelerator that can efficiently implement these reduced operations is, on the other hand, certainly possible [1-3]**. In fact, **in neuromorphic accelerators such as Loihi [4], FP multiplications are typically avoided using message passing between processors that model multiple neurons**. This point has now been clarified in the Subsection 'Inference Efficiency' in Section 4. Lastly, we would like to highlight that the premise of energy efficiency in almost all existing SNN works is based on this avoidance of FP multiplications using binary activations.
> >
> > [Missing $\lambda_H$]
> >
> > We have now added $\lambda_H$ in all the correct places in Section 3.2.
> >
> > [Reproducibility]
> >
> > We have now uploaded our code in the supplementary materials zip file. We commit to releasing our code to the public upon acceptance. Our hyperparameter details are provided in Appendix A2.
> >
> > [Concerns regarding novelty]
> >
> > Please refer to the 'Concerns regarding novelty' section of the 'Response to all reviewers' comment above. To the best of our knowledge, of this **our work is the first to jointly optimize the distribution of the SNN membrane potential and the relative placement of the SNN threshold** to improve the trade-off between test accuracy and the number of time steps. You are right that the loss of accuracy due to binary activations is obvious, however, in this paper, we study how we can mitigate this loss compared to SOTA full-precision DNNs. Our results demonstrate that our approach is superior compared to both multi-time-step SNN models as well as BNN models in terms of accuracy-efficiency trade-off.

---

> > ### Comment · Reviewer_YzPx · 2022-11-23
> > **Confusing answers**
> >
> > First, the term "time-independent variant of the popular leaky integrate and fire" on Page 2 is very confusing. The potential in LIF should evolve over time because of the leakage as well as spike inputs distributed over time. I am very puzzled about this term.
> >
> > $\textbf{Unclear settings for learning on event-data}$: the authors report the accuracy on DVS-CIFAR10 in Appendix. The authors should clarify (i) how the input spikes distributed over time are integrated to output the membrane potential and (ii) how the above-threshold potential is reset. I wonder if the authors did not consider the integration over time so that the SNN does not learn any dynamic features of the event-data instead it learns static features of snapshots of event-data at given timesteps. Please clarify this.
> >
> > $\textbf{Unclear hardware under consideration}$: the authors considers dedicated event-based hardware such as neuromorphic hardware. Otherwise, the improvement on spike sparsity is meaningless since it can't improve CE in any sense. In this regard, it is odd to compare CE of SNNs with DNN (based on synchronous layer-wise MACs) when addresing the contribution of neuron-wise operations like comparisons to the total CE. I believe that the larger the sparsity to more the contribution of neuron-wise operations to the total CE.

---

> > > ### Author Response · Authors · 2022-11-24
> > > **Response to follow-up concerns [1/2]**
> > >
> > > Dear reviewer YzPx,
> > >
> > > Thank you for your response and discussing with us. We apologize for the confusion, and we have tried our best to clarify your concerns below.
> > >
> > > [Imprecise Sentence]
> > >
> > > We would replace that sentence in Page 2 as "**In this work, we propose one-time-step SNNs where, unlike the traditional LIF model, the membrane potential of each neuron do not integrate and leak over time, as illustrated in Eq. 2**."
> > >
> > > [Unclear settings for learning on event-data]
> > >
> > > We apologize that we did not elaborate on our SNN setup for the evaluation on the DVS-CIFAR10 dataset shown in Appendix A.4 and the multiple time steps shown in Appendix A.3. For both these experiments, we **use the traditional LIF model** for the neurons where **the membrane potential integrates the weight modulated input spikes and leaks over time** (except the experiment on one time step where such integration is not possible).
> > >
> > > We use the **soft reset mechanism that reduces the membrane potential by the threshold value** when an output spike generated. It has been shown that soft reset minimizes the information loss by allowing the spiking neuron to carry forward the surplus potential above the firing threshold to the subsequent time step [1,2].
> > >
> > > **We use our proposed combination of Hoyer regularized training and Hoyer spike layer to estimate the per layer threshold, while we train the weights and leak term using SGL**.
> > >
> > > The equations governing our LIF model for our multi-time-step SNNs on static and DVS datasets are shown below.
> > >
> > > $U_i^{temp}(t)=\lambda U_i(t-1)+\sum_j W_{ij}{S_j(t)}$
> > > $S_i(t) = 1 \ \text{if } U_i^{temp}(t)>V^{th} \ \text{else} \ 0$
> > >
> > > $U_i(t) = U_i^{temp}(t)-S_i(t)V^{th}$
> > >
> > > Here $U_i(t)$ denotes the membrane potential of the $i^{th}$ neuron at timer step $t$, $S_i(t)$ denotes the binary output of the $i^{th}$ neuron at time step $t$, $V^{th}$ denotes the threshold, $\lambda$ denotes the leak, and $W_{ij}$ denotes the weight connecting the pre-synaptic neuron $j$ and the neuron $i$.
> > >
> > > We will add these details in the revised manuscript.
> > >
> > > [Unclear hardware under consideration]
> > >
> > > Comment: *the authors considers dedicated event-based hardware such as neuromorphic hardware. Otherwise, the improvement on spike sparsity is meaningless since it can't improve CE in any sense*.
> > >
> > > Response: We agree that dedicated event-driven hardware might be the most suitable to leverage the energy efficiency of SNNs coming from the spike sparsity. That said, the spike sparsity can also leveraged in synchronous deep learning accelerators developed for DNNs, such as tensor processing units (TPU) [3], Eyeriss [4], etc. with the support of zero-gating logic [5]. The hardware overhead for this zero-gating logic is negligible compared to the large sparsity of SNNs [5], especially our one-time-step SNNs.
> > >
> > > Comment: *In this regard, it is odd to compare CE of SNNs with DNN (based on synchronous layer-wise MACs) when addresing the contribution of neuron-wise operations like comparisons to the total CE*.
> > >
> > > Response: We understand your confusion here. While evaluating the CE, we have considered the most suitable hardware for each of SNN and DNN. For SNNs, this is dedicated event-driven hardware (or synchronous Von-Neumann hardware with zero-gating logic that incurs negligible sparsity overhead as described above) that can leverage the energy efficiency coming from both sparsity and accumulate-only operations. For DNNs, this is synchronous Von-Neumann hardware. Otherwise, it is not possible to leverage the benefits of the SNN energy efficiency as you alluded to in your response.
> > >
> > > That said, we understand a holistic comparison of the CE of SNNs and DNNs would heavily depend on the architectural dataflow and the underlying implementation of each hardware, and hence, the comparisons that we provide may be approximate. However, this is the best comparison that we can provide without a detailed model of the hardware, which has also been used in a lot of prior SNN works [6-8].
> > >
> > > We would also like to highlight that our primary goal is to compare the CE of our one-time-step SNNs with existing multi-time-step SNNs (please see Fig. 3c), and not DNNs. This comparison can be made more accurately compared to DNNs, and we show significant compute efficiency improvements on dedicated event-driven hardware in Fig. 3c . We would clarify these points in the revised version.

---

> > > > ### Author Response · Authors · 2022-11-24
> > > > **Response to follow-up concerns [2/2]**
> > > >
> > > > Comment: *I believe that the larger the sparsity to more the contribution of neuron-wise operations to the total CE.*
> > > >
> > > > Response: You are correct! However, with the spiking activities of our one-time-step SNN model, **the total number of neuron-wise comparison operations still contributes negligibly (${\sim}$0.32\%) to the total CE**. We show the layer wise spiking activities (which is equal to (100 - sparsity) in percentage) of our network with VGG16 on CIFAR10 below. We also show the layer wise compute energies coming from the accumulate operations and the neuron-wise comparison operations separately.
> > > >
> > > > The compute energy due to accumulate operations in a conv. or fully-connected layer $l$, except the first layer, is denoted as
> > > >
> > > > $CE_{acc}^l=S^l\times DNN_{flops}^l\times 0.9$ pJ
> > > >
> > > > where $S^l$ and $DNN_{flops}^l$ denote the spiking activity and number of floating point operations without considering the sparsity respectively. Note that $0.9$ pJ is the energy consumed by each accumulate operation. The first layer is direct encoded, has no sparsity, and involves FP MAC operations which consume $4.6$ pJ for each operation. Hence, the compute energy for the first layer is denoted as
> > > >
> > > > $CE_{acc}^1=DNN_{flops}^1\times 4.6$ pJ
> > > >
> > > > In contrast, the compute energy due to neuron-wise comparison operations for layer $l$ that follows a conv. or fully-connected layer is denoted as
> > > >
> > > > $CE_{com}^l=DNN_{com}^l\times 0.7$ pJ
> > > >
> > > > where $DNN_{com}^l$ denotes the total number of comparison operations at layer $l$, and each comparison operation consumes $0.7$ pJ.
> > > >
> > > > | **Layer** | **Input Spiking Activity** | **Accumulate CE ($\mu$J)** | **Comparison CE ($\mu$J)** | **Total CE ($\mu$J)** |
> > > > |-----------|------------------|--------------|-----------|------------|
> > > > | 1       | 1.0             | 8.14      | 0.04     | 8.20        |
> > > > | 2       | 0.33              | 18.91     | 0.04     | 18.95        |
> > > > | 3       | 0.23             | 3.59      | 0.09     | 3.68        |
> > > > | 4       | 0.17         | 3.93       | 0.09     | 4.02         |
> > > > | 5       | 0.17 | 7.85        |   0.18  |   8.03       |
> > > > | 6 | 0.19       | 19.62     | 0.18| 19.80        |
> > > > | 7 | 0.13       | 9.19      | 0.18 | 9.37          |
> > > > | 8 | 0.07       | 5.33       | 0.37| 5.70          |
> > > > | 9 | 0.12       | 31.31      | 0.37 | 31.68          |
> > > > | 10 | 0.06      | 7.83       | 0.37 | 8.20          |
> > > > | 11 | 0.09       | 17.61      | 0.37 | 17.98          |
> > > > | 12 | 0.36      | 281.79       | 0.37 | 282.16          |
> > > > | 13 | 0.5       | 543.58       | 0.37 | 543.95         |
> > > > | 14 | 0.16       | 0.19       | 0.002 | 0.19          |
> > > > | 15 | 0.14       | 0.29       | 0.002 | 0.19          |
> > > > | 16 | 0.31      | 0.03       | 0.00 | 0.03          |
> > > >
> > > > | **Avg. Spiking Activity** | **Total Accumulate CE ($\mu$J)**         | **Total Comparison CE ($\mu$J)** |
> > > > |-----------|------------------|--------------|
> > > > | 0.22      | 959.19            | 3.06      |
> > > >
> > > >
> > > > This trend is observed in the existing low-time-step SNNs as well. For example, DIET-SNN [9] has a spiking activity of 0.086 per time step (total of 0.43 across 5 time steps) on average with VGG16 on CIFar10. This also translates to a negligible contribution (${\sim}$0.81\%) of the neuron-wise operations to the total CE.
> > > >
> > > > For the neuron-wise operations to contribute equally with the accumulate operations, the spiking activity needs to be 0.0007 per time step, which is too low to train an accurate SNN. We will add these discussions to the revised version of the paper.
> > > >
> > > > References
> > > >
> > > > [1] B. Han et al. "RMP-SNN: Residual Membrane Potential Neuron for Enabling Deeper
> > > > High-Accuracy and Low-Latency Spiking Neural Network", CVPR 2020
> > > >
> > > > [2] N. Rathi et al. "Enabling Deep Spiking Neural Networks with Hybrid Conversion and Spike Timing Dependent Backpropagation", ICLR 2020
> > > >
> > > > [3] N. P. Jouppi et al. "In-Datacenter Performance Analysis of a Tensor Processing Unit", ISCA 2017
> > > >
> > > > [4] Y. Chen et al. "Eyeriss: An Energy-Efficient Reconfigurable Accelerator for Deep Convolutional Neural Networks", IEEE JSSC 2017
> > > >
> > > > [5] P. K. Chundi et al. "Always-On Sub-Microwatt Spiking Neural Network Based on Spike-Driven Clock- and Power-Gating for an Ultra-Low-Power Intelligent Device", Frontiers in Neuroscience 2021
> > > >
> > > > [6] S. Kundu et al. "Spike-Thrift: Towards Energy-Efficient Deep Spiking Neural Networks by Limiting Spiking Activity via Attention-Guided Compression", WACV 2021
> > > >
> > > > [7] P. Panda et al. "Towards Scalable, Efficient and Accurate Deep Spiking Neural Networks with Backward Residual Connections, Stochastic Softmax and Hybridization", Frontiers in Neuroscience 2020
> > > >
> > > > [8] I. Garg et al. "DCT-SNN: Using DCT to Distribute Spatial Information over Time for
> > > > Low-Latency Spiking Neural Networks", ICCV 2021
> > > >
> > > > [9] N. Rathi et al. "DIET-SNN: Direct Input Encoding With Leakage and Threshold Optimization in Deep Spiking Neural Networks",  TNNLS 2021

---

> > > > > ### Author Response · Authors · 2022-11-30
> > > > > **Gentle reminder**
> > > > >
> > > > > Dear reviewer YzPx,
> > > > >
> > > > > We were wondering if our response addressed your concerns regarding our confusing answers. We would be more than happy to discuss with you if you need further clarifications! Thanks for your time and patience.
> > > > >
> > > > > Regards,
> > > > >
> > > > > Authors

---

### Official Review · Reviewer_qbCE · 2022-10-24

**Confidence:** 4
**Correctness:** 3
**Technical Novelty And Significance:** 3
**Empirical Novelty And Significance:** 3
**Recommendation:** 8

**Clarity, Quality, Novelty And Reproducibility:**

* Will the authors share their code?

* Eq 1 is misleading, there is only one type step here.

* |u_l| is described in the text before it's actually used in the equations

* what are the blue areas on Fig 1?

* what is the dataset in Table 1?

* https://arxiv.org/abs/2102.04159 and https://arxiv.org/abs/2007.05785 should be included in Table 4



**Strength And Weaknesses:**

STRENGTHS:

The accuracy they reach with only one timestep is impressive, better than the previous state-of-the-art.

WEAKNESSES:

The theory is unclear:

* "we estimate the value of the Hoyer extremum as He(ul) ="
I think what follows is not the value of H but the value of the element of u_l at the extremum.

* a SNN with one timestep is actually a somewhat degenerated case because there is no temporal integration. It boils down to a vanilla feed-forward artificial neural network, with Heaviside as the activation function. This also corresponds to the first neuron model proposed by McCulloch and Pitts and 1943, also known as threshold gates. This should be discussed.

* according to Table 5 what matters is the Hoyer spike, not the Hoyer regularization. This raises a question: could Hoyer spike be used alone, without the Hoyer regularization?

* does the approach extend to multistep SNNs?



**Summary Of The Paper:**

The authors propose a new method to train spiking neural networks (SNNs) with only one timestep. The method is based on a new regularization term ("Hoyer"), and a new threshold adjustment method ("Hoyer spike", Eq 4).



**Summary Of The Review:**

A potentially promising method, but some aspects are unclear.

---

> ### Author Response · Authors · 2022-11-17
> **Response to reviewer qbCE**
>
> Thanks for your valuable comments and suggestions to improve the quality of our work. Please see our response below and the revised version of the manuscript.
>
> [Unclear theory regarding Hoyer extremum]
>
> The value of the element of the membrane potential u_l at the extremum is actually called the Hoyer extremum $H_e(u_l)$, which is different from the Hoyer regularizer $H(u_l)$. To avoid any confusion, we have now modified the notation of the Hoyer extremum to $Ext(u_l)$.
>
>
> [Discussion on one-time-step SNNs]
>
> We acknowledge your point and have discussed this point in Section 1 of our paper. Please refer to the 'Concerns regarding lack of bio-plausibility/temporal dynamics of one-time-step SNNs' section of the 'Response to all reviewers' comment for a more holistic discussion on this point.
>
> [Effect of Hoyer spike without Hoyer regularizer]
>
> We have now run an ablation study on the effect of the Hoyer spike alone without the Hoyer regularizer on our training framework. Our results are added in Table 5. As we can see, the Hoyer spike is more effective compared to Hoyer regularizer and leads to a 2.37\% (0.33\%) increase in test accuracy with VGG16 (ResNet18) on CIFAR10. The Hoyer regularizer alone contributes to 0.21\% (0.20\%) increase in accuracy with VGG16 (ResNet18). All these results are reported as the mean from five runs with distinct seeds.
>
>
> [Extension to multi-step SNNs]
>
> We have now added results of multi-time-step SNNs with our proposed training framework in Table 8 of Appendix A.3 and below. As we can see, the test accuracy increases with the number of time steps (till a total of 4 time steps on CIFAR10 with VGG16 beyond which the accuracy saturates), however, the spiking activity increases significantly, thereby increasing the compute energy and the temporal "overhead" increases, thereby increasing the memory cost due to the repetitive access of the membrane potential and weights across the different time steps.
>
> | **Architecture** | **Time steps** | **Acc. (%)** | **Spiking Activity (%)** |
> |------------------|----------------|--------------|------------------|
> | VGG16            | 1              | 93.44        | 21.87            |
> | VGG16            | 2              | 93.71        | 44.06            |
> | VGG16            | 4              | 94.14        | 74.88            |
> | VGG16            | 6              | 94.04        | 102.33           |
>
> [Reproducibility]
>
> We have now uploaded our code in the supplementary materials zip file. We commit to releasing our code to the public upon acceptance.
>
> [Clarity]
>
> 1. We have now removed Eq. 1 from Section 2.1 of the paper, acknowledging that it may mislead the readers.
>
> 2. We have now described |u_l| in the correct position.
>
> 3. The blue areas in Fig. 1 denote the frequency distribution of the clipped and normalized membrane potential ($z_l^{clip}$ illustrated in Eq. 4) of our one-time-step SNNs. This distribution is not-to-scale as it is overlaid with the activation function of our SNN.
>
> 4. The dataset in Table 1 is CIFAR10 which has now been added to the caption of the table.
>
> [Comparison with other works]
>
> We have now compared with [1] and [2] in Table 2 of the updated paper.
>
> References
>
> [1] Fang, Wei et al. "Deep Residual Learning in Spiking Neural Networks", NeurIPS 2021
>
> [2] Fang, Wei et al. "Incorporating Learnable Membrane Time Constant to Enhance Learning of Spiking Neural Networks", ICCV 2021

---

> > ### Comment · Reviewer_qbCE · 2022-11-18
> > **Concerns addressed, thanks!**
> >
> > All my concerns have been addressed, thanks!
> > I've raised my rating to 8.

---

### Official Review · Reviewer_ujG1 · 2022-10-25

**Confidence:** 3
**Correctness:** 3
**Technical Novelty And Significance:** 3
**Empirical Novelty And Significance:** 3
**Recommendation:** 5

**Clarity, Quality, Novelty And Reproducibility:**

The paper is clear. The quality is fair. The novelty is limited. There is no code provided, so it is not reproducible.

**Strength And Weaknesses:**

Strength

1. The paper is well-written and easy to follow.
2. The experiments show the method with low-latency.

Weaknesses
1. The problem importance is unclear. One-time-step SNNs is simply binary network. The lack of dynamics make it less biologically plausible.
2. Limited novelty. The proposed method is simply training with HOYER regularizer, while similar method has been proposed in [1]


[1]Huanrui Yang, Wei Wen, and Hai Li. Deephoyer: Learning sparser neural network with differentiable scale-invariant sparsity measures. In International Conference on Learning Representa- tions, 2020.

**Summary Of The Paper:**

The paper presents a training framework (from scratch) for one- time-step SNNs that uses a novel variant of the recently proposed Hoyer regularizer. By estimating the threshold of each SNN layer as the Hoyer extremum of a clipped version of its activation map, the approach not only downscales the value of the trainable threshold, but also shifts the membrane potential values away from the threshold. The approach outperforms existing spiking, binary, and adder neural networks in terms of accuracy-FLOPs trade-off for complex image recognition tasks.


%%%%%%%%%%%%%

Update:
The authors have addressed part of my initial concern about bio-plausibility and limitation of BNN. Yet, I don't think the proposed method is that novel. So I change my score to 5.

**Summary Of The Review:**

The paper fail to clarify the problem importance, and the proposed method is limited novel. So I vote for a reject.

---

> ### Author Response · Authors · 2022-11-17
> **Response to reviewer ujG1**
>
> Thanks for your valuable comments and suggestions to improve the quality of our work. Please see our response below and the revised version of the manuscript.
>
> [Concerns regarding bio-plausibility]
>
> Please refer to the 'Concerns regarding lack of bio-plausibility/temporal dynamics of one-time-step SNNs' section of the 'Response to all reviewers' comment above. In particular, our problem statement is the **development of accurate one-time-step SNN models which avoid any temporal "overhead" and yields significant improvement in energy and latency efficiency** compared to existing works. We believe that **such a problem statement is extremely relevant and timely**, given the increased deployment of energy-efficient computer vision models in resource-constrained edge devices. **Reducing the number of SNN time steps for static vision tasks has also been a popular problem statement** in the recent past. Here, we provide a non-exhaustive list of papers that are focused on this problem statement and published in top-tier ML conferences and journals in the recent past [1-4]. In fact, upon a quick glance on the current ICLR submissions on SNNs, we could find at least a couple of papers [5-6] studying the same problem.
>
> [Limited Novelty]
>
> We humbly disagree that our work is limited novel. Please refer to the 'Concerns regarding novelty' section of the 'Response to all reviewers' comment above. To the best of our knowledge, our work is the **first to jointly optimize the distribution of the SNN membrane potential and the relative placement of the SNN threshold** to improve the trade-off between test accuracy and the number of time steps. In particular, we significantly surpass existing iso-architecture multi-time-step SNNs and binary neural networks (BNNs) in terms of accuracy and energy efficiency. We would also like to clarify that **our proposed method is not about simply training with Hoyer regularizer**. One **key novelty of our approach is our proposed Hoyer spike layer** that leverages the distribution shift of the activation map enabled by Hoyer regularizer by setting the IF threshold to the Hoyer extremum. This cleanly maps the activations to binary outputs, reducing noise and improving training convergence. In fact, as shown from our ablation studies in Table 5, **our Hoyer extremum-based spike layer is more effective than the Hoyer regularizer which further justifies our novelty**.
>
> [Reproducibility]
>
> We have now uploaded our code in the supplementary materials zip file. We commit to releasing our code to the public upon acceptance.
>
> References
>
> [1] Li, Yuhang et al. "A Free Lunch From ANN: Towards Efficient, Accurate Spiking Neural Networks Calibration", ICML 2021
>
> [2] Deng, Shikuang et al. "Optimal Conversion of Conventional Artificial Neural Networks to Spiking Neural Networks", ICLR 2021
>
> [3] Rathi, Nitin et al. "DIET-SNN: A low-latency spiking neural network with direct input encoding and leakage and threshold optimization", TNNLS 2021
>
> [4] Kim, Youngeun et al. "Neural architecture search for spiking neural networks", ECCV 2022
>
> [5] "A unified optimization framework of ANN-SNN Conversion: towards optimal mapping from activation values to firing rates", submitted to ICLR 2023
>
> [6] "Spike Calibration: Bridging the Gap between ANNs and SNNs in ANN-SNN Conversion", submitted to ICLR 2023

---

> > ### Comment · Reviewer_ujG1 · 2022-11-19
> > **Thanks for your response.**
> >
> > 1.Concerns regarding bio-plausibility. I am still concerning about the bio-plausibility of the paper. The authors claim to avoid overhead computation in SNN. But the time latency can be reduced with a small number step SNN, like 5-step SNN. Furthermore,  I don't see a discussion the capacity between "BNN" and SNN, as SNN is proved to be "universal".
> >
> >
> > 2. Limited novelty. I agree the proposed Hoyer layer is somewhat novel, but I don't see the motivation of "jointly optimize the relative placement of the SNN threshold".

---

### Official Review · Reviewer_pFTU · 2022-11-01

**Confidence:** 4
**Correctness:** 3
**Technical Novelty And Significance:** 2
**Empirical Novelty And Significance:** 3
**Recommendation:** 5

**Clarity, Quality, Novelty And Reproducibility:**

Clarity & Quality: The introduction of the method details is clear, but there lacks a reasonable explanation for the motivation and effects of the method. Also, there are some questions regarding the experimental results and comparisons. (see above weakness)

Novelty: The Hoyer regularizer is introduced by previous works. The Hoyer spiking layer is new.

Reproducibility: Good.


**Strength And Weaknesses:**

Strengths:

1. This paper conducts extensive experiments on static image tasks, including classification and detection, with different network structures, and considers many aspects such as accuracy, energy efficiency, and training and inference time. The effect of quantization of weights is also considered.

2. This paper considers comparisons with AddNNs and BNNs and reports promising results.

Weakness:

1. The “one-time-step SNN” in the paper is actually an ANN with binary activations. It does not consider temporal dynamics of neuron models, so it should not belong to “spatio-temporal computing paradigm” as mentioned in the first sentence of the abstract. It cannot handle temporal inputs such as speech or DVS data, which may possibly be more suitable for SNN with temporal processing ability. So this paper is more close to binary or quantized ANNs compared with general SNNs. The introduction, related work, and comparisons should have more focus on those ANNs than SNNs, and some introduction such as the dynamics of the LIF neuron model in Section 2.1 almost has no relation with models in this paper.

2. The motivation and the effect to introduce Hoyer regularizer and Hoyer spike layer are not clear enough. As introduced in Section 2.2, Hoyer regularizer is first introduced to induce sparsity. Then adding this regularizer for membrane potentials will encourage membrane potentials towards 0, implying that spikes are encouraged to be sparser. However, in Section 3.1, it is said that “it is crucial to reduce the value of the threshold to generate enough spikes for better network convergence”, which is *contradictory* to the regularizer that encourages fewer spikes. Why to combine these two opposite components and why it can work? Besides, there is no explanation for why “Hoyer extremum” is introduced to adjust the threshold and why it can work to balance the threshold and include proper data distribution as shown in Figure 1(a), why Eq. (4) considers clip function, why combining this “Hoyer extremum” with the trainable threshold can be better than the single trainable threshold? There lack significant explanations for the methods.

3. Some descriptions are imprecise. For example, in Section 3.1, “This is because if a neuron does not emit a spike, the pre-synaptic weight connected to it cannot be updated”. What does “pre-synaptic weight” mean? I suppose the authors mean that the synaptic weight will not be updated if the pre-synaptic neuron does not emit a spike when gradients of $w_{ij}$ from neuron $i$ to $j$ are calculated by $g_{u_j}*o_i$, where $g_{u_j}$ is the gradient for $u_j$.
And in Section 3.3, “Note that unlike existing multi-time-step SNN architectures that avoid BN due to difficulty in convergence”. This is not up-to-date since most recent SNN works use BN to achieve high performance [1,2,3,4].

4. The energy results in Table 6 are strange. As introduced in the “Effect on Quantization” paragraph, transforming the full-precision ACs to 2-6 bit ACs will lead to 4.8-15.2 reduction in compute energy. Why in Table 6, there are more than 400-1000 times reduction? Not to mention that only convolutional layers are quantized as said in the paragraph.

5. The comparison with BNN and AddNN in Table 7 is not clear enough. The reported accuracy of the method is based on full-precision (also note that the full-precision results in Table 6 is inconsistent with Table 4 and Table 7) but the reported energy is based on 2-bits quantization, which is inconsistent. And how is the energy of BNN and AddNN calculated? Is AddNN also quantized? As for the proposed methods, even with quantization, it is unclear to me why the energy can be much lower than BNN with only pop-count operations. I think pop-count operations are much more efficient than ACs. These details should be reported and explained.

6. The reference style in the paper is with poor readability. The authors had better distinguish parenthetical and narrative types, i.e. \citep and \citet in the latex.

[1] Zheng, H., Wu, Y., Deng, L., Hu, Y., & Li, G. (2021). Going deeper with directly-trained larger spiking neural networks. In Proceedings of the AAAI Conference on Artificial Intelligence (Vol. 35, No. 12, pp. 11062-11070).

[2] Li, Y., Guo, Y., Zhang, S., Deng, S., Hai, Y., & Gu, S. (2021). Differentiable spike: Rethinking gradient-descent for training spiking neural networks. Advances in Neural Information Processing Systems, 34, 23426-23439.

[3] Deng, S., Li, Y., Zhang, S., & Gu, S. (2021). Temporal Efficient Training of Spiking Neural Network via Gradient Re-weighting. In International Conference on Learning Representations.

[4] Meng, Q., Xiao, M., Yan, S., Wang, Y., Lin, Z., & Luo, Z. Q. (2022). Training High-Performance Low-Latency Spiking Neural Networks by Differentiation on Spike Representation. In Proceedings of the IEEE/CVF Conference on Computer Vision and Pattern Recognition (pp. 12444-12453).

**Summary Of The Paper:**

This paper proposes to train one-time-step SNNs with a Hoyer regularizer and Hoyer spike layer. The proposed Hoyer spike layer uses adaptive threshold based on the Hoyer extremum of membrane potentials, and a Hoyer regularization for membrane potentials is added in the loss function. Experiments on static image classification and object detection tasks demonstrate competitive performance and energy efficiency.

**Summary Of The Review:**

In summary, this paper lacks some significant explanations for the methods despite the good performance, and there are some questions about the experiments that should be tackled. Besides, the paper should have more focus on the more similar binary or quantized neural networks.

---

> ### Author Response · Authors · 2022-11-17
> **Response to Reviewer pFTU [1/3]**
>
> Thanks for your valuable comments and suggestions to improve the quality of our work. Please see our response below and the revised version of the manuscript.
>
> [Lack of temporal dynamics and similarity with BNNs]
>
> Please refer to the 'Concerns regarding lack of bio-plausibility/temporal dynamics of one-time-step SNNs' section of the 'Response to all reviewers' comment above. We would like to highlight that our one-time-step SNN models are structurally similar to the sparsity-induced binary unipolar neural networks (BNN) [1] that have '0' and '1' as the two states (and not the more popular bi-polar BNNs that have '1' and '-1' as the two states). These sparsity-induced BNNs offer significantly higher sparsity than bi-polar BNNs, which leads to significantly fewer pop-count operations. However, these networks are usually hard to optimize, and as shown in Table 7, lead to lower test accuracy than our proposed approach. This shows the efficacy of our approach even from the perspective of BNNs.
>
> Moreover, we should emphasize that due to the similarity with the SOTA sparsity-induced BNNs [1], our Hoyer regularized training approach, coupled with the Hoyer spike layer, can be applied to SOTA sparsity-induced BNNs [1]. Moreover, as illustrated in our response to all reviewers, our proposed approach can be extended to multiple time steps, thereby leading to a small but significant accuracy increase (see Appendix A.3) at the cost of a significant increase in memory and compute costs. Hence, **our approach applies to the continuum of models between BNNs and low-time-step SNNs and can be seen as bridging the gap between BNN and SNN communities**.
>
> [Lack of motivation & explanation of proposed approach]
>
> In contrast to L1 and L2 regularizers, Hoyer regularized training increases the sparsity of the activation maps at the cost of retaining larger activation values. This characteristic stems from its derivative whose zero point is a scalar known as the "Hoyer extremum". When an element of the activation map tensor is smaller than Hoyer extremum, it is pushed toward 0. When it is larger than the Hoyer extremum, it is pushed away from 0. Hence, though our Hoyer regularized training increases the sparsity (intended for energy efficiency in SNNs), the threshold estimation (Hoyer extremum of a clipped version of the activation map as shown in Eq. 4) via our novel Hoyer spike layer ensures a relatively clean separation of the activation map to binary ('0' or '1') outputs.
>
> Note that our IF threshold value is indeed less than the trainable threshold (clipped value of the activation map as shown in Eq. 4) value used in existing works as shown in Appendix A1. Had we used the larger clipping threshold to further encourage more sparsity, we would significantly compromise the test accuracy as shown in Table 5. Please check the 2.63\% (90.45\% vs 93.13\%) accuracy difference for VGG16 and the 0.53% accuracy difference for ResNet18 (90.95\% vs 91.48\%). This may be due to the fewer  spikes emitted by the pre-synaptic neuron which leads to fewer updates of the synaptic weight and/or the imprecise separation of the activation map distribution to binary outputs. In summary, **we optimize the trade-off between accuracy and spiking activity for one-time-step SNNs** via our proposed training framework.
>
> [Why is clip function used?]
>
> The clip function is used to **get rid of a few outlier values in the activation map that may otherwise unnecessarily increase the value of the Hoyer extremum, i.e., threshold value**, thereby reducing the accuracy, without any noticeable increase in energy efficiency. In fact, the test accuracy with VGG16 on CIFAR10 drops by more than 1.4\% (from 93.13\% obtained by our training framework) to 91.7\% without the clip function.
>
> [Why combining this “Hoyer extremum” with the trainable threshold can be better than the single trainable threshold?]
>
> Since existing works use surrogate gradients (and not real gradients) to update the thresholds with appropriate initializations, it is difficult to estimate the optimal value of the IF thresholds. On the other hand, our Hoyer extremums dynamically change with the activation maps particularly during the early stages of training (coupled with the distribution shift enabled by Hoyer regularized training), which enables our Hoyer extremum-based scaled thresholds to be closer to optimal. In fact, as shown from our ablation studies in Table 5, **our Hoyer extremum-based spike layer is more effective than the Hoyer regularizer** which further justifies the importance of the combination of the Hoyer extremum with the trainable threshold.

---

> > ### Author Response · Authors · 2022-11-17
> > **Response to reviewer pFTU [2/3]**
> >
> > [More focus on binary networks]
> >
> > We have rewritten Section 1 (Introduction & Related work) by adding details of binary neural networks (particularly the sparsity-induced ones that are similar to our one-time-step SNNs) and removing unnecessary details on multi-time-step SNNs. We have also removed the dynamics of the LIF neuron model from Section 2.1 of the paper, acknowledging that this may mislead the readers. Finally, we have added further analysis of the accuracy-compute trade-off of AddNNs and BNNs in the 'Comparison with AddNNs and BNNs' subsection of Section 4.
> >
> > [Imprecise descriptions]
> >
> > We have now modified the first statement as "This is because if a pre-synaptic neuron does not emit a spike, the synaptic weight connected to it cannot be updated as its gradient from neuron i to j is calculated by $g_{u_j}{\times}o_i$, where $g_{u_j}$ is the gradient of the membrane potential $u_j$ and $o_i$ is the output of the neuron i". We have also modified the second statement as "Similar to recently developed multi-time-step SNN models [1-4], our models show that BN helps increase the test accuracy with one time step.".
> >
> > [Clarifications on energy benefits from quantization]
> >
> > We apologize for this mistake. We have now corrected the value of the compute energy of our one-time-step SNNs with floating point weights in Table 6. With this corrected value, the weight quantization now leads to a 4.8-13.8$\times$ reduction in compute energy as illustrated in the Subsection 'Effect on Quantization' in Section 4.
> >
> > [Clarifications on comparisons with BNNs and AddNNs]
> >
> > We have now consistently reported the compute energy and accuracy corresponding to the bit-precision of the weights of our one-time-step SNNs in Table 7. As we can see, our one-time-step SNNs with 2-bit weights yield higher test accuracy compared to iso-architecture BNNs with lower energy consumption. Compared to iso-architecture AddNNs, our one-time-step SNNs consume 5.5$\times$ less energy with only a 0.26\% drop in accuracy on CIFAR10. When the weights in the SOTA AddNN are quantized, for example to 2-bit precision, our one-time-step SNNs improve both the test accuracy and energy efficiency as shown in Table 7 and below. All these results demonstrate the superiority of our one-time-step SNNs compared to BNNs and AddNNs in terms of accuracy-efficiency trade-off.
> >
> > | **Reference**             | **Architecture** | **Dateset** | **Acc. (%)** | **CE(J)** |
> > |---------------------------|------------------|-------------|--------------|-----------|
> > | [3]                       | BNNs             | CIFAR10     | 89.6         | 0.022     |
> > | [1]                       | BNNs             | CIFAR10     | 90.2         | 0.019     |
> > | [1]                       | BNNs             | ImageNet    | 59.7         | 3.6       |
> > | [4]                       | BNNs             | CIFAR10     | 91.9         | 0.073     |
> > | [5] (FP weights)          | AddNNs           | CIFAR10     | 93.72        | 1.62      |
> > | [5] (2-bit weights)       | AddNNs           | CIFAR10     | 92.08        | 0.12      |
> > | [5] (FP weights)          | AddNNs           | ImageNet    | 67.0         | 77.8      |
> > | [6] (FP weights)          | AddNNs           | CIFAR10     | 91.56        | 1.62      |
> > | This work (FP weights)    | SNNs             | CIFAR10     | 93.44        | 0.297     |
> > | This work (2-bit weights) | SNNs             | CIFAR10     | 92.34        | 0.021     |
> > | This work (FP weights)    | SNNs             | ImageNet    | 68.00        | 14.28     |
> >
> > [How are compute energies of BNNs and AddNNs calculated? Is AddNN quantized?]
> >
> > The compute energy of BNNs and AddNNs are estimated using the energy consumed by pop-count and AC operations respectively, followed by analytical computation of the total FLOPs as shown in Eq. 14. The pop-count and AC energies (and MAC energies for comparison with non-spiking DNNs) are estimated from post place-and-route FPGA simulations on the Kintex-7 platform using custom RTL specifications. As shown in the updated Table 7, we use quantization-aware training approach (same as used in our one-time-step SNNs in Table 6) to quantize AddNNs, which leads to reduction in the compute cost at the expense of 1.64\% drop in accuracy on CIFAR10.
> >
> > [Inconsistent full-precision results in different tables]
> >
> > Table 4 and 7 compares our one-time-step SNNs with existing works in SNNs and BNNs respectively. Hence, the accuracy numbers in these tables correspond to our best results after 600 epochs of training, and for fairness, we report the best results from the existing works as well. However, Table 6 shows ablation results on the effect of the quantization of weights in our one-time-step SNNs, and hence we do not strive for the best results here. Instead, for training efficiency, we report the test accuracies of the four models after 300 epochs of training. We have clarified this discrepancy in the Subsection 'Effect on Quantization' in Section 4.

---

> > > ### Author Response · Authors · 2022-11-17
> > > **Response to reviewer pFTU [3/3]**
> > >
> > > [Why do one-time-step SNNs consume lower energy compared to BNNs]
> > >
> > > Our one-time-step SNNs yield spiking activities of around 22\%, which leads to fewer compute operations compared to bi-polar BNNs [2]. Thus, although the pop-count operation consumes 20\% lower energy compared to 2-bit AC operations, the significantly high sparsity in our one-time-step SNNs lead to higher compute efficiency compared to bi-polar BNNs. On the other hand, our one-time-step SNNs consume similar energy compared to unipolar BNNs [1] while yielding 3.2\% higher accuracy on CIFAR10 at iso-architecture. The energy consumption is similar because the 20\% advantage of the pop-count operations is mitigated by the 22.5\% lower sparsity of the unipolar BNNs [1] compared to our one-time-step SNNs.
> > >
> > > [Concerns regarding readability]
> > >
> > > We have now used '\citep' instead of '\cite' to improve the readability of our paper.
> > >
> > > References
> > >
> > > [1] Wang, Peisang et al. "Sparsity-Inducing Binarized Neural Networks", AAAI 2020
> > >
> > > [2] Liu, Zechun et al. "ReActNet: Towards Precise Binary Neural Network with Generalized Activation Functions", ECCV 2020
> > >
> > > [3] Charbel Sakr et al. " True gradient-based training of deep binary activated neural networks via continuous binarization", ICASSP 2018
> > >
> > > [4] James Diffenderfer and Bhavya Kailkhura. "Multi-prize lottery ticket hypothesis: Finding accurate binary neural networks by pruning a randomly weighted network", ICLR 2021
> > >
> > > [5] Hanting Chen et al. "Addernet: Do we really need multiplications in deep learning?", CVPR 2020
> > >
> > > [6] Wenshuo Li et al. "Winograd Algorithm for AdderNet", PMLR 2021

---

> ### Comment · Reviewer_pFTU · 2022-11-18
> **Thank you for the response and further questions**
>
> I appreciate the authors for their great efforts on the response. Most of the questions are clarified, and many experiments are supplemented to support good empirical contributions, so I raise my score to 5. Still, I have some questions about the explanation of the proposed approach.
>
> In the response, the authors explain their combination of Hoyer regularization and Hoyer spike layer as optimizing the trade-off between high sparsity and high spiking activity. However, I do not see the support for this claim. In Table 5, adding Hoyer regularization will actually increase the firing activity for VGG-16, and simply adding Hoyer spike layer for ResNet18 does not increase the activity. There is, however, an opposite phenomenon for the other structure. And there is no illustration if the trade-off can be controlled, e.g. by adjusting the coefficient of Hoyer regularization. It is not clear if the above phenomenon is due to randomness, so I would suggest the authors report results based on multiple runs and observe if there is indeed a common phenomenon. Otherwise, the explanation (and the motivation) for the method does not accord with the empirical results.

---

### Official Review · Reviewer_hEby · 2022-11-01

**Confidence:** 4
**Correctness:** 3
**Technical Novelty And Significance:** 2
**Empirical Novelty And Significance:** 2
**Recommendation:** 5

**Clarity, Quality, Novelty And Reproducibility:**

he contributions are clear and the results are good. I am not sure about novelty since it is a mix of methods that have existed in SNN/ANN literature and putting together seem to make the model better. -I am concerned about whether this is truly advanategous SNN framework as I have raised questions around SNN implementation and the sparsity in weakness.

**Strength And Weaknesses:**

Very comprehensive results

Simple yet effective idea

Since the authors use a regularization technique, I am wondering if the authors can shed light on how their method differs from previous temporal BN methods proposed by prior works that have shown accuracy improvement while decreasing the total timesteps [1, 2].

[1]Kim, Y., & Panda, P. (2020). Revisiting batch normalization for training low-latency deep spiking neural networks from scratch. Frontiers in neuroscience, 1638.

[2] Zheng, Hanle, et al. "Going deeper with directly-trained larger spiking neural networks." Proceedings of the AAAI Conference on Artificial Intelligence. Vol. 35. No. 12. 2021.

There is a plethora of works today on SNN algorithmic training- precisely talking about how we can get improved accuracy with less timesteps. But, I am more concerned by the fact that in such large-scale settings, are SNNs going to be actually advantageous? The authors show some simplistic energy estimation results which is grossly approximate. For true energy estimation, they have to consider memory access and data access energy which turns out to expend a lot of computations in SNNs given their repeated time-wise computation. In a recent work [3], true energy estimation on a systolic accelerator precisely shows that SNNs are not very advantageous over ANNs because repeated timestep computations will lead to redundant access of weights and membrane potentials is going to further add to energy unless we really improve the sparsity rate. Can the authors kindly comment on this - and it may be worthwhile for authors to include a discussion n the relevance of using more mainstream tools for energy estimation rather than just doing analytical modeling of FLOPS count?

 [3] Yin, Ruokai, et al. "SATA: Sparsity-Aware Training Accelerator for Spiking Neural Networks." arXiv preprint arXiv:2204.05422 (2022).

Coming to my next point, there is a recent work [4] that explores sparse SNNs using lottery ticket hypothesis or NAS [R5]  to truly take advantage of SNNs energy efficiency over ANNs. Can the authors comment on how their model in terms of parameter count compares to these sparse SNN models which in fact show SOTA accuracy on CIFAR10,100 with very low timestep count?

[4] Kim, Youngeun, et al. "Lottery Ticket Hypothesis for Spiking Neural Networks." arXiv preprint arXiv:2207.01382 (2022).

[R5] Kim, Youngeun, et al. "Neural architecture search for spiking neural networks." arXiv preprint arXiv:2201.10355 (2022).

Finally, I think it is well known that SNNs will be more suited to DVS or event based tasks as compared to standard digital camera recognition models. Recent works have shown superiority of SNNs over ANNs on these neuromorphic datasets [5, 6, 7]. Can the authors run their model on one of these datasets and compare to [5,6,7]?

[5] Li, Yuhang, et al. "Neuromorphic Data Augmentation for Training Spiking Neural Networks." arXiv preprint arXiv:2203.06145 (2022).

[6] Kim, Youngeun, and Priyadarshini Panda. "Optimizing deeper spiking neural networks for dynamic vision sensing." Neural Networks 144 (2021): 686-698.

[7] Kim, Y., Chough, J., & Panda, P. (2022). Beyond classification: directly training spiking neural networks for semantic segmentation. Neuromorphic Computing and Engineering.

**Summary Of The Paper:**

The paper presents an interesting training algorithm for training SNNs from scratch with only one timestep.

**Summary Of The Review:**

I am not very convinced by the novelty and true efficiency advantage of this method since many comparisons and experiments are limited.

---

> ### Author Response · Authors · 2022-11-17
> **Response to Reviewer hEby**
>
> Thanks for your valuable comments and suggestions to improve the quality of our work. Please see our response below and the revised version of the manuscript.
>
> [Comparison with temporal BN methods]
>
> Our proposed Hoyer regularized training yields one-time-step SNNs which **avoid any temporal overhead** and results in high energy efficiency. On the other hand, previous efforts on temporal BN methods [1,2] for SNNs incur multiple time steps (2-30), while still yielding lower test accuracy compared to our approach. These BN approaches leverage the temporal dynamics in rate/direct-coded SNNs and hence are somewhat orthogonal to our work. Note that we have added these references in Section 1 of the paper.
>
> [More realistic energy estimation methodology]
>
> We agree with the reviewer that our analytical modeling of the FLOP counts do not accurately reflect the total energy consumption of our SNN models since the memory costs should be considered (depending on the dataflow and underlying hardware implementation). However, we would like to highlight that our **one-step-SNNs avoid the repetitive read and write accesses of the weights and membrane potential** that are typical in multi-time-step SNNs. As you have correctly pointed out, these repetitive and redundant memory accesses significantly adds to the energy, as shown in the energy estimation on a systolic array accelerator [3]. Our SNNs incur a significantly lower memory footprint compared to these multi-time-step SNNs. Based on the improved energy evaluation framework proposed in [3] which models the memory and sparsity cost along with the compute cost, **our models are significantly more energy-efficient** (2.1-14.7$\times$ reduction in total energy from 1.4-3.5$\times$ reported in the original paper compared to multi-time-step SNNs). We have updated Fig. 3 and the subsection 'Inference Efficacy' of Section 4 to reflect these additional improvements.
>
> [Comparison with lottery ticket hypothesis & NAS-based approaches]
>
> Lottery ticket hypothesis (LTH) and NAS-based approaches in the SNN domain can yield low-latency SNN models with significant improvements in energy efficiency. While the LTH work [4] attempts to optimize the "temporal" overhead by iterative time-step pruning, the NAS work [5] leverages the temporal information in multi-time-step SNNs to design optimal architectures with backward connections. Hence, our one-time-step SNN work is somewhat orthogonal to both of these approaches, however, these works have been discussed in Section 1 of the paper.
>
> [Evaluation on DVS tasks]
>
> The inherent temporal dynamics in SNNs may be better leveraged in DVS or event-based tasks [6-9] compared to standard static vision tasks that are studied in this work.
> Upon your request, we have now evaluated our framework on the DVS-CIFAR10 dataset, which provides each label with only 0.9k training samples, and is considered the most challenging event-based dataset [6]. **We surpass the test accuracy of existing works [6-8] by 1.30\%** on average at iso-time-step as shown in Table 9 in Appendix A.4 and below. In fact, **our accuracy gain is more significant at low time steps**, thereby implying the portability of our approach to DVS tasks. Note that similar to static datasets, a large number of time steps increase the temporal overhead in SNNs, resulting in a large memory footprint and spiking activity. We have added these results in Appendix A.4.
>
> | **Reference** | **Training**         | **Architecture** | **Acc. (%)**  | **Time steps** |
> |-----------|------------------|--------------|-----------|------------|
> | [6]       | TET              | VGGSNN       | 83.17     | 10         |
> | [6]       | TET              | VGGSNN       | 75.20     | 4          |
> | [6]       | TET              | VGGSNN       | 68.12     | 1          |
> | [7]       | tdBN+NDA         | VGG11        | 81.7      | 10         |
> | [8]       | SALT+Switched BN | VGG16        | 67.1      | 20         |
> | This work | Hoyer reg.       | VGGSNN       | **83.68** | 10         |
> | This work | Hoyer reg.       | VGGSNN       | **76.17** | 4          |
> | This work | Hoyer reg.       | VGGSNN       | **69.80** | 1          |
>
> [Concerns regarding novelty]
>
> We humbly disagree that our work is a mix of methods that have existed in the ANN/SNN literature. We have clarified the novelty of our work in the 'Concerns regarding novelty' section of the 'Response to all reviewers' comment above. To the best of our knowledge, our work is the first to jointly optimize the distribution of the SNN membrane potential and the relative placement of the SNN threshold to improve the trade-off between test accuracy and the number of time steps.

---

> > ### Author Response · Authors · 2022-11-17
> > **References**
> >
> > [1] Kim, Y., & Panda, P. (2020). Revisiting batch normalization for training low-latency deep spiking neural networks from scratch. Frontiers in neuroscience, 1638.
> >
> > [2] Zheng, Hanle, et al. "Going deeper with directly-trained larger spiking neural networks." Proceedings of the AAAI Conference on Artificial Intelligence. Vol. 35. No. 12. 2021.
> >
> > [3] Yin, Ruokai, et al. "SATA: Sparsity-Aware Training Accelerator for Spiking Neural Networks." arXiv preprint arXiv:2204.05422 (2022).
> >
> > [4] Kim, Youngeun, et al. "Lottery Ticket Hypothesis for Spiking Neural Networks." arXiv preprint arXiv:2207.01382 (2022).
> >
> > [5] Kim, Youngeun, et al. "Neural architecture search for spiking neural networks." arXiv preprint arXiv:2201.10355 (2022).
> >
> > [6] Deng, Shikuang et al. "Temporal Efficient Training of Spiking Neural Network via Gradient Re-weighting", ICLR 2022
> >
> > [7] Li, Yuhang, et al. "Neuromorphic Data Augmentation for Training Spiking Neural Networks." arXiv preprint arXiv:2203.06145 (2022).
> >
> > [8] Kim, Youngeun, and Priyadarshini Panda. "Optimizing deeper spiking neural networks for dynamic vision sensing." Neural Networks 144 (2021): 686-698.
> >
> > [9] Kim, Y., Chough, J., & Panda, P. (2022). Beyond classification: directly training spiking neural networks for semantic segmentation. Neuromorphic Computing and Engineering.

---

> > > ### Comment · Reviewer_hEby · 2022-11-20
> > > **Response to Authors**
> > >
> > > Thanks to the authors for providing a detailed rebuttal response and a comprehensive comparative analysis. I understand that there is an advantage to the Hoyer regularization aspect and in that you reduce the overall latency and get good accuracy results. But, I still would like to understand how this makes it novel. In my humble opinion, the SNN that the authors have developed is a BNN-like model. But, making it more BNN-like will evidently reduce the precision cost and therefore allows energy efficiency. I agree the authors have SOTA results, but it's because they use HOYER techinque proposed  in previous ANN works and threshold balancing/placement technique proposed in some previous SNN works. I still think the authors have provided a combination of different techniques to get better results and the techniques in themselves are not novel. Therefore, I would like to keep my rating as is - 5:marginally below acceptance.

---

### Author Response · Authors · 2022-11-17
**Response to all reviewers [1/2]**

We thank all the reviewers for their detailed constructive feedback and we have provided individualized responses to all of them. Here, however, we would like to clarify two concerns that have been raised by multiple reviewers.

### [Concerns regarding novelty]

In this work, we propose a Hoyer Regularizer and a Hoyer spike layer to jointly optimize the distribution of the SNN membrane potential and the placement of the SNN threshold in order to improve the accuracy-energy trade-off.

Our Hoyer regularized training involves normalizing the activation map using a learned clipping threshold followed by clipping to one. This training acts to shift the distribution of the normalized activation map away from its Hoyer extremum, as shown in the blue boxes in Fig. 1a.

Our proposed **Hoyer spike layer leverages this activation distribution shift by setting the IF threshold to the Hoyer extremum**. This cleanly maps the activations to binary outputs, reducing noise and improving training convergence. The clipping of the activation map also helps to reduce the value of our IF threshold (the Hoyer extremum) compared to existing works [1-4] that either initialize this value from pre-trained ANN distributions (for ANN-to-SNN conversion approaches) or optimize this value using surrogate gradient learning [5-6]. Our reduced IF thresholds improve both accuracy and energy efficiency, as quantified in this paper.

Existing works based on ANN-to-SNN conversion optimize the pre-trained ANN distribution and/or the error between the ANN and SNN pre-activation map to improve the trade-off between the number of time steps and the test accuracy of the SNN. Other SNN works that are based on surrogate gradient learning tend to optimize the nature and shape of the non-differentiable gradient and the loss function.

In contrast, to the best of our knowledge, **no work (even in sparsity-induced binary neural networks (BNN) that are similar to our one-time-step SNNs) has jointly optimized the distribution of the SNN membrane potential (or activation map in the context of BNNs) and the relative placement of the SNN (or BNN) threshold to improve the accuracy-energy trade-off**.

This feature is the key to achieving SOTA accuracy in one time step.

### [Concerns regarding lack of bio-pluasibility/temporal dynamics of one-time-step SNNs]

SNNs have been extensively used as energy-efficient ML models at the edge for generic static computer vision tasks in the recent past due to their sparsity and use of AC-only operations. However, the early exploration of SNN models required hundreds of time steps [1-2] which significantly increased the latency (encoding a static image across all the time steps) and greatly offset the energy advantages of sparsity and AC-only operations. Since then, there have been significant advances in the SNN community [3-8] that reduced the required number of time steps, including ANN-to-SNN conversion, surrogate gradient learning (SGL), lottery ticket hypothesis, and neural architecture search. These low time-step SNN models reduce the overall latency and the spiking activity, which leads to more sparsity and hence, lower energy consumption. One time step is the lowest latency one can obtain with SNNs which **avoids any temporal "overhead"** in classifying/detecting the inputs. In particular, one-time-step SNNs **avoid the repetitive read and write accesses of the membrane potential** that are typical in multi-time-step SNNs. Coupled with the reduced latency and memory footprint, **one-time-step SNNs also reduce the spiking activity which leads to lower energy consumption** (see Fig. 3(c) for a thorough comparison of energy with multi-step SNNs). The key empirical result of this paper is that we achieve all these savings without almost any drop in test accuracy compared to the SOTA multi-time-step SNNs and even the sparsity-induced binary neural networks which are structurally similar to the baseline one-time-step SNNs (with thresholds and weights as trainable parameters). Moreover, our proposed approach can be extended to multiple time steps, thereby leading to small but significant accuracy increase (see Appendix A.3) at the cost of significant increase in memory and compute cost. Hence, our approach acts as a continuum between one time-step sparsity-induced BNNs and low time-step SNNs, and can help bridge both the BNN and SNN community for static image recognition and object detection tasks. [Continued in next comment]

---

> ### Author Response · Authors · 2022-11-17
> **Response to all reviewers [2/2]**
>
> We would also like to point out that **in the vast majority of these static tasks, there is no temporal dynamics in the associated dataset**. Rather, the static image pixel values from these datasets are typically encoded across multiple time steps to better approximate the full-precision activation map in the inner SNN layers. Hence, **the temporal dynamics is artificially injected into the SNN as part of the input encoding**. Moreover, some forms of input encoding, such as direct encoding [9], incur no loss of information at  one time step. Hence, we firmly believe that the lack of temporal dynamics is actually an advantage, rather than a weakness for these recognition and detection applications.
>
> In contrast, we acknowledge the lack of temporal dynamics of one-time-step SNN models in naturally temporal data, such as those involving event-driven cameras. Upon the request of Reviewer hEby, we have now evaluated our framework on the DVS-CIFAR10 dataset as shown in Table 9 of Appendix A.4. We surpass the test accuracy of existing DVS works by 1.30\% on average at iso-time-step and architecture. In fact, **our accuracy gain is more significant at low time steps, thereby implying the portability of our approach to DVS tasks**. Note that similar to static datasets, a large number of time steps in DVS datasets increase the temporal overhead in SNNs, resulting in a large memory footprint and spiking activity. Hence, a small number of time steps may still be warranted for DVS tasks.
>
> References:
>
> [1] A. Sengupta et al. "Going Deeper in Spiking Neural Networks: VGG and Residual Architectures." Frontiers in Neuroscience 2019
>
> [2] Y. Cao et al. "Spiking deep convolutional neural networks for energy-efficient object recognition." IJCV 2015
>
> [3] S. Deng et al. "Optimal Conversion of Conventional Artificial Neural Networks to Spiking Neural Networks." ICLR 2021
>
> [4] T. Bu et al. "Optimal ANN-SNN Conversion for High-accuracy and Ultra-low-latency Spiking Neural Networks." ICLR 2022
>
> [5] Y. Li et al. "Differentiable Spike: Rethinking Gradient-Descent for Training Spiking Neural Networks." NeurIPS 2021
>
> [6] S. Deng et al. "Temporal Efficient Training of Spiking Neural Network via Gradient Re-weighting." ICLR 2022
>
> [7] Y. Kim et al. "Lottery Ticket Hypothesis for Spiking Neural Networks." ECCV 2022
>
> [8] Y. Kim et al. "Neural architecture search for spiking neural networks." ECCV 2022
>
> [9] N. Rathi et al. "DIET-SNN: Direct Input Encoding With Leakage and Threshold Optimization in Deep Spiking Neural Networks." TNNLS 2021

---

### Author Response · Authors · 2022-11-17
**Looking forward to further discussions!**

Dear Reviewers,

We sincerely hope that our response and the revised manuscript have addressed all your concerns. We have highlighted our changes in the manuscript. In particular, we believe we have clarified the two key concerns regarding novelty and lack of temporal dynamics in our one-time-step SNNs. Based on your suggestions, we have also conducted multiple experiments highlighted below. We believe that these experiments further demonstrate the efficacy of our approach.

1. Evaluation of our training framework on multi-time-step SNNs (see Table 8 in Appendix)
2. Extension of our approach to DVS tasks (see Table 9 in Appendix)
3. Further analysis on the trade-off between accuracy, bit-precision, and compute energy of AddNNs, BNNs, and our one-time-step SNNs (see Table 7)
4. Further ablation studies on the use of clip function and Hoyer regularizer (see Table 5 and Appendix A.5)

We would appreciate it if you could let us know whether you have any other questions or suggestions. We are looking forward to discussions that can further improve our work. Thanks!

Best regards,

The Authors

---

### Comment · Area_Chair_zFLm · 2022-11-18
**Please respond to author rebuttals**

Dear Reviewers,

The authors have submitted their rebuttals. Please have a look and respond to their efforts. This will be a respect to their hard work. Many thanks!

Area Chair

---

### Public Comment · ~Justin_Liu1 · 2022-11-18
**Notable points of concerns regarding the claims of the paper - 1**

Thanks to the authors and reviewers for presenting and evaluating this interesting work. In this regard we would like to mention few point (some of them are already mentioned by the respected reviewers as well) that may be important for the proper evaluation of the paper.

`1. On contribution: Concern with motivation and novelty as highlighted by respected reviewers`

**Motivation**. The Introduction inducts few BNN papers, however, did not incorporate the recent BNN variants [1-3]. This to our opinion question the motivation to have an alternative that indeed isn't outperforming the BNNs. Moreover, the training is significantly costlier due to general requirements of SNNs to have more epochs as opposed to DNNs.

**Novelty**. The paper heavily relies on existing work of Hoyer regularizer and trainable thresholding to present their work. The learnable aspect of the "clipping threshold" as claimed in the rebuttal as a part of novelty, is not, the clipping aspect was introduced here following the criteria of Hoyer extremum. On the contrary earlier works used train threshold and often used to apply a constant clipping factor.

**So, to our understanding, the precise difference is in the "aspect" of "how the clipping of trainable threshold is done"**

Also, it should be noted that the clipped threshold being lower than $v^{th}$, comes as an inherent property due to the selection of the Hoyer square reg. based normalization. This has already been shown in one of the papers referred by the authors as an inherent Hoyer reg. property [5] (see Eq. 1-3). Moreover, as the improvement is negligible, a constant clipping factor might have a good job as well.

**post-rebuttal comment**. We would also like to highlight one the authors' rebuttal response on novelty: "*In contrast, to the best of our knowledge, no work (even in sparsity-induced binary neural networks (BNN) that are similar to our one-time-step SNNs) has jointly optimized the distribution of the SNN membrane potential (or activation map in the context of BNNs) and the relative placement of the SNN (or BNN) threshold to improve the accuracy-energy trade-off.*"

* The recent most BNNs leverage +1/-1 activation thus there is **no real need** of including this study of Hoyer. Both the [0,1] BNN and SNN (this paper) suffers from significant accuracy drop compared to the other alternative while having significant compute burden due to larger model/bit-width overhead.

* The SOTA BNN models are **better in accuracy and energy efficient** compared to the presented SNN counter parts.

* Improvement of sparse activation is largely a function of architecture change that is already proposed in the traditional BNNs. Interestingly, **for ResNet18 the Hoyer reg. effectively reduces sparsity** (Table 5 of the paper). So, the **results contradicts with the claim of Hoyer consistently inducing sparsity**.

* It is not clear why this is a  "joint optimization". The authors learn a threshold $v^{th}$ same as earlier works (that is a part of **single optimization associated to SGD/ADAM optimizer driven loss including Hoyer reg. component to minimize**), scales them with Hoyer squared regularize (channel wise) as computed via activation maps (no-trainable parameters here). likewise, BN learnable parameters are not a part of joint opt either, this is not correct claim.

Also, the architectural modifications adapted in contrast to existing one-time step approach is not new, the authors themselves highlighted this.

`Thus as highlighted by respected reviewers, the paper may still miss the motivation and novelty considering the post rebuttal comments/draft`.

`2. On results: Ambiguous claiming on being better than BNN variants, which as per published works is not the case`

We see the authors justified the motivation of 1-step SNN due to their potential benefits over BNNs (state-of-the-arts). We think this is not the case. Follows is the comparison with published 1-bit weight-activation networks that are significantly less computing burden than the VGG16, yet achieve better results than this works.

On CIFAR-10

| Model | Method | W/A bit width|  Accuracy |
|--|--|--|--|
|ResNet18 | ReCU (2021) [1] | 1/1 | **92.8%** |
|ResNet18 | AdaBin (2022) [2] | 1/1 | **93.1%** |
| VGG16| This paper| 2/1 | 92.34% |
| ResNet18 | This paper| Floating point | 91.48% |
| ResNet18 | This paper| 2/1 | Not provided |

On ImageNet

| Model | Method | W/A bit width|  Accuracy |
|--|--|--|--|
|ResNet18 | ReActNet (2020) [3]| 1/1 | 65.5% |
|ResNet18 | AdaBin (2022) [2] | 1/1 | 66.4% |
|ReActNet-A| ReActNet (2020) [3] | 1/1 | **69.4%** |
|ReActNet-A| AdaBin (2022) [3] | 1/1 | **70.4%** |
| ResNet50 | This paper| FP weights | 66.32% |
| VGG16| This paper| FP weights | 68% |
| VGG16| This paper| 2/1 | Not provided |

*[These show that several times even the FP-32 SNNs can't outperform the binary DNN counterparts]*

---

> ### Public Comment · ~Justin_Liu1 · 2022-11-18
> **Notable points part - 2**
>
> all the models used in the mentioned BNN methods often **require orders of less memory and FLOPs due to fewer parameters**. Also, note these works ensured both weights and activations are +1/-1 allowing XNOR and pop-count to happen. Also, you may see Section 3.4 of the AdaBin paper [3] to  have more clarity on how the computation cost can be significantly less.
>
> **Thus, we would encourage the authors to clearly seen why should the approach of 2-bit quantized SNN be more beneficial compared to the efficient counterparts of BNNs that need much less parameters to reach >4% accuracy benefit.  This makes it virtually not possible to compensate orders of high compute cost of quantized SNN (as opposed to low model and 1/1 SOTA BNNs) extract benefits of computation with only 22% additional activation sparsity. Thus we believe a major part of the manuscript's claim is misleading or unjustified or not true.**
>
> We are also not sure why the authors mentioned about significant activation sparsity, as the results show only 22% act sparsity that can often reduce compared to the baseline 27->22% for similar architecture ResNet.
>
> **3. On results: This paper does not really outperform the 1 time step SNN** Though authors claimed they "significantly" outperform SOTA SNNs, we would point out that the results do not clearly outperform the 1-time SNN found in the literature [4].On ImageNet
>
> | Model | Method | t step |  Accuracy |
> |--|--|--|--|
> |VGG16 | IIR-SNN [4] | 1 | **69%** |
> |VGG16 | This paper | 1 | 68% |
> | ResNet50 | This paper| 1 | 66.32% |
>
> We thus would urge the concerned evaluators and authors to seriously reconsider the claims made in this paper.
>
> **4. On results: "Several" ambiguous comparisons in the updated draft as well** Though the authors  show many results, however, most of the comparisons are unfair and not exactly comparable, thus creates confusion as a scientific paper. Example, Table 3 has comparisons of the SNN backbone with BNNs, models of different architecture. *We would also like to highlight the ReActNet is a binary variant of ResNet18, that the authors compared with FP-32 weight SNN with ResNet50 backbone!!* We are not sure what comprehensive message we can infer. As we have already highlighted the SNNs' performance are not at all comparable to that of currently released BNN variants.
>
>  It is also not clear how 1.25x inference speed up comes compared to (Chowdhury et al., 2021)? ideally the proposed block has additional layers, that should increased the inference time.
>
> **Nevertheless, as the comparison should actually be with BNN alternatives, it would be more fare to have a comparison with training/inference time of BNNs. Also, SNNs takes significantly more training epochs to converge. Authors' Fig. 4 actually justifies the improved training and inference of conventional DNN even for 1 epoch. The proposed SNN training is significantly more carbon emitting compared to alternative of DNN/BNNs.**
>
> **5. On results: On DVS dataset** As with 1 tstep the results on DVS is significantly poor, and as the accuracy is of equivalent BNNs are significantly high, this again demotivates the use of the proposed work compared to BNN alternatives, as now it should be applied to DVS to replace 1 step SNN. This further weakens the contribution of this work.
>
> *This makes most of the empirical evaluations and comparisons either incomplete or unfair*.
>
> We hope this should help the paper's comprehensive evaluation. Following are the references for consideration by the authors and reviewers to better evaluate the paper.
>
>
> [1] Xu et al., ReCU: Reviving the Dead Weights in Binary Neural Networks, ICCV 2021.[2] Liu et al., ReActNet: Towards Precise Binary Neural Network with Generalized Activation Functions, ECCV 2020.[3] Tu et al., AdaBin: Improving Binary Neural Networks with Adaptive Binary Sets, ECCV 2022.[4] Chowdhury et al., Towards Ultra Low Latency Spiking Neural Networks for Vision and Sequential Tasks Using Temporal Pruning, ECCV 2022.[5] Yang et al., "DeepHoyer: Learning Sparser Neural Networks with Differentiable Scale-Invariant Sparsity Measures" ICLR 2020.

---

> > ### Author Response · Authors · 2022-11-20
> > **Irresponsible and misleading claims [1/4]**
> >
> > Thanks for reading our paper, and providing your detailed feedback. We believe most (if not all) of the points you raised are either misleading or incorrect. We have provided a point-by-point response to each of your comments below.
> >
> > [Motivation: Comparison with BNNs]
> >
> > We have never claimed in our motivation that we aim to propose an alternative to BNNs or improve the accuracy of existing BNNs. Instead, we primarily compare with sparsity-induced uni-polar BNNs due to the similarity in their structure, sparsity, and lack of temporal dynamics with our proposed one-time-step SNNs.
> >
> > Moreover, unlike your claim, we would like to emphasize that **our training times are not significantly higher than those for DNNs since we train with only one time-step from scratch (see Fig. 4). Our training time should be even smaller than SOTA BNNs [1-3] which often require iterative pre-training**.
> >
> > In particular, we see **bi-polar BNNs in a different class of solutions because they do not yield any sparsity and thus lead to 4-5$\times$ more compute than our one-time-step SNNs** on average (thanks to our 20-25\% spiking activity). In addition to the increased compute, **we were unable to find any bi-polar BNNs that surpass the accuracy of our one-time-step SNNs (even with 2-bit weights) without significant architectural changes that introduce a large number of non-binary operations. Specifically, the two references [2-3] that you have claimed achieve SOTA results are ReactNet-based which incur custom non-linear functions (such as RPReLU that are significantly more complex compared to threshold/ReLU operations), duplicated basic blocks that lead to ${\sim}$2$\times$ increase in the FLOPs and parameter count, floating point operations in skip connections, etc.** We are also surprised that **you have omitted our best results from your table, and only included those models that do not surpass the accuracy of the SOTA BNNs**!
> >
> > [Novelty]
> >
> > From your comments regarding the novelty of our paper, we do not believe you have characterized our claims correctly and appear to have missed our novelty. We strongly recommend you to carefully read Section 3.1 of the paper and the 'Concerns regarding novelty' section here (https://openreview.net/forum?id=0L8tuglXJaW&noteId=aQLzlDL2NCH).
> >
> > Comment: *Also, it should be noted that the clipped threshold being lower than $v^{th}$, comes as an inherent property due to the selection of the Hoyer square reg. based normalization. This has already been shown in one of the papers referred by the authors as an inherent Hoyer reg. property [5] (see Eq. 1-3).*
> >
> > Response: For your kind information, the clipped threshold is equal to (and not lower than) $v^{th}$. Assuming that you meant the Hoyer extremum based IF threshold instead of the clipping threshold, we want to highlight that **the threshold downscaling is not due to an inherent property of the Hoyer regularizer. It is precisely due to the clipping and normalization of the membrane potential, with a novel proof shown in Appendix A.1**. None of the Eq. 1-3 in [5] can downscale the threshold without the clipping and normalization.
> >
> > Comment: *Moreover, as the improvement is negligible, a constant clipping factor might have a good job as well.*
> >
> > Response: This is again another misleading and irresponsible claim. A constant clipping factor implies that the threshold is not trainable. **It is extremely difficult to obtain close to SOTA accuracy in one-time-step without training the threshold, as evidenced from our own experience and the fact there are no published works available demonstrating otherwise**.
> >
> > Comment: *The recent most BNNs leverage +1/-1 activation thus there is no real need of including this study of Hoyer.*
> >
> > Response: We have nowhere claimed that the bi-polar BNNs need to include Hoyer regularizer to improve their performance, and that is also not our goal. Our claim only applies to uni-polar BNNs which need a threshold to yield binary (0/1) activations, similar to our SNNs.

---

> > > ### Author Response · Authors · 2022-11-20
> > > **Irresponsible and misleading claims [2/4]**
> > >
> > > Comment: *Both the [0,1] BNN and SNN (this paper) suffers from significant accuracy drop compared to the other alternative while having significant compute burden due to larger model/bit-width overhead.*
> > >
> > > Response: This is again unfortunately incorrect. There is no evidence our one-time-step SNNs suffer from a significant accuracy drop compared to other alternatives. **Even without the sparsity advantage and with additional non-binary computations/increased FLOPs and parameters, the SOTA BNNs cannot surpass the test accuracy of our one-time-step SNNs on CIFAR10**. From another perspective, **our SNNs with 2-bit FP weights yield more than 3.4$\times$ reduction in compute energy compared to SOTA BNNs at iso-architecture without any evidence of a significant drop in accuracy**. On ImageNet, we do see a $2.4\%$ accuracy drop in our one-time-step SNNs compared to SOTA ReactNet-based AdaBin [3], however, the comparison is not only at iso-architecture, but also involves significantly more compute and parameters as explained above.  Moreover, the arxiv version of Adabin [3] was released 41 days before the ICLR deadline, and hence, can be regarded as contemporary work. **It is your referenced BNNs [2-3] that incur a significant compute burden compared to SNNs (not the other way around) at iso-architecture due to their lack of sparsity, additional non-binary operations, increased FLOPs and parameters due to duplicated basic blocks and expensive iterative training**!
> > >
> > > Comment: *The SOTA BNN models are better in accuracy and energy efficient compared to the presented SNN counter parts.*
> > >
> > > Response: As mentioned above, SOTA BNN models are better in accuracy in ImageNet (and not CIFAR) compared to our one-time-step SNNs at the cost of significant non-binary computations, architectural changes inducing ${\sim}$2$\times$ more FLOPs and parameters and training complexity. Additionally, we think it is completely unfair to compare the accuracy and energy efficiency with different architectural configurations. It is thus unfair to compare the SOTA BNN on ResNet with our SOTA results on VGG. It has been shown in multiple SNN works that, unlike traditional DNNs and BNNs, SNNs yield better test accuracy with VGG compared to ResNet.
> > >
> > > Comment: *Improvement of sparse activation is largely a function of architecture change that is already proposed in the traditional BNNs.*
> > >
> > > Response: We do not understand what your point is here. We wonder how traditional bi-polar BNNs, especially the ones you highlighted, can yield sparsity. The papers you cited [1-3] have no mention of introducing sparsity. Additionally, it is unfair to compare BNN and SNN sparsity across different architectural configurations.
> > >
> > > Comment: *Interestingly, for ResNet18 the Hoyer reg. effectively reduces sparsity (Table 5 of the paper). So, the results contradicts with the claim of Hoyer consistently inducing sparsity.*
> > >
> > > Response: See our response [Lack of motivation & explanation of proposed approach] to reviewer pFTU. We have already clarified this discrepancy of inducing sparsity enabled by Hoyer regularizer and increasing spiking activity enabled by Hoyer spike layer.
> > >
> > > Comment: *It is not clear why this is a  "joint optimization". The authors learn a threshold  same as earlier works (that is a part of single optimization associated to SGD/ADAM optimizer driven loss including Hoyer reg. component to minimize), scales them with Hoyer squared regularize (channel wise) as computed via activation maps (no-trainable parameters here). likewise, BN learnable parameters are not a part of joint opt either, this is not correct claim.*
> > >
> > > Response: As we described in our rebuttal to the reviewers, **the joint optimization is in the distribution of the SNN membrane potential (enabled by Hoyer regularizer) and the estimation of the SNN threshold (enabled by Hoyer spike layer). The optimization is joint because they are coupled during training. We do not scale with the Hoyer squared regularizer, but rather the Hoyer extremum of the clipped membrane potential, which is layer-wise (and not channel-wise).** This Hoyer extremum dynamically tunes the threshold based on the value of the membrane potential, which we show, provides higher test accuracy compared to a single trainable threshold. Lastly, **we have never claimed BN learnable parameters are a part of joint optimization. We said that we calculate the exponential average of the Hoyer extremums during training (a phenomenon that is similar to BN), and use the same during inference.**

---

> > > > ### Author Response · Authors · 2022-11-20
> > > > **Irresponsible and misleading claims [3/4]**
> > > >
> > > > [Results]
> > > >
> > > > Comment: *All the models used in the mentioned BNN methods often require orders of less memory and FLOPs due to fewer parameters.*
> > > >
> > > > Response: We do not understand the basis of your argument here. The memory size i.e., the parameter count is an architectural trait that is independent of our (or of any of the existing SNNs) training methodology. **We have demonstrated we incur less FLOPs compared to existing bi-polar BNNs at iso-architecture. If we include your referenced SOTA BNNs [2,3] for comparison, the gap would grow even further since they are significantly more compute and parameter expensive as explained above.** Since BNNs have been around for a longer time than low-time-step SNNs, it is quite natural that BNNs give superior accuracy for efficient backbones such as MobileNet, ResNet, etc. Hence, it is unfair to compare the FLOPs of SNNs with parameter-inefficient backbones such as VGG against BNNs with parameter-efficient backbones such as ResNet. That said, we would add these comparisons with SOTA BNNs in our future version. Our future work includes improving the accuracy of our one-time-step SNNs with parameter-efficient backbones such as ResNet, MobileNet, etc.
> > > >
> > > > Comment: *We would encourage the authors to clearly seen why should the approach of 2-bit quantized SNN be more beneficial compared to the efficient counterparts of BNNs that need much less parameters to reach >4% accuracy benefit.*
> > > >
> > > > Response: We completely disagree with this assessment. The >4% accuracy benefit that you quote corresponds to cherry-picked results taking the best BNN with significant increase in FLOPs/parameters and non-binary computations vs our worse results!
> > > >
> > > > Comment: *This makes it virtually not possible to compensate orders of high compute cost of quantized SNN (as opposed to low model and 1/1 SOTA BNNs) extract benefits of computation with only 22% additional activation sparsity.*
> > > >
> > > > Response: **We do not think the term "additional" activation sparsity is accurate because SOTA BNNs do not have any sparsity. Moreover, the 22% spiking activity corresponds to 4.8x reduction in FLOPs, which implies a similar factor of reduction in compute cost.**
> > > >
> > > > Comment: *Thus we believe a major part of the manuscript's claim is misleading or unjustified or not true.*
> > > >
> > > > Response: **We completely disagree with this accusation.** We would like to kindly ask you which part of the manuscript is "misleading" or "unjustified" or "not true"? All of our claims are backed by sufficient data, and our codes are currently made available which can help anyone reproduce our results. **We strongly recommend that you retract this statement.**
> > > >
> > > > Comment: *We are also not sure why the authors mentioned about significant activation sparsity, as the results show only 22% act sparsity that can often reduce compared to the baseline 27->22% for similar architecture ResNet.*
> > > >
> > > > Response: **"Sparsity" should be read as "spiking actitivity" which had been corrected in the revised version of the paper. 22-25\% spiking activity corresponds to 4-4.8$\times$ reduction in compute cost which is indeed "significant".**
> > > >
> > > > [Comparison with SOTA one-time-step SNN]
> > > >
> > > > Your claim that we do not outperform existing SNNs at iso-time-step and iso-architecture is irresponsible and misleading. The reference [4] brought up first came online on 3rd November, while the ICLR submission deadline was 29th September. We compared our results with the arxiv version of [4] (https://arxiv.org/pdf/2110.05929.pdf) which came online around one year ago and yielded 67.71% accuracy, 0.29% lower compared to us. Nevertheless, even without considering this accuracy drop, [4] incurs significantly more training time at iso-architecture compared to us due to its iterative nature of training. This has been clearly highlighted in the subsection 'Training & Inference Time Requirements' of Section 4 and Fig. 4.
> > > >
> > > > Comment: *It is also not clear how 1.25x inference speed up comes compared to (Chowdhury et al., 2021)? ideally the proposed block has additional layers, that should increased the inference time.*
> > > >
> > > > Response: It seems like you again misunderstand our approach. We do not have any additional layers. The Hoyer spike layer is basically an IF layer with the threshold estimated as the Hoyer extremum of the clipped and normalized membrane potential. **The 1.25$\times$ inference speed up is due to more efficient PyTorch tensor operations used in our code which may be better optimized using the underlying CUDA compiler.**

---

> > > > > ### Author Response · Authors · 2022-11-20
> > > > > **Irresponsible and misleading claims [4/4]**
> > > > >
> > > > > Comment: *Nevertheless, as the comparison should actually be with BNN alternatives, it would be more fair to compare with training/inference time of BNNs. Also, SNNs takes significantly more training epochs to converge. Authors' Fig. 4 actually justifies the improved training and inference of conventional DNN even for 1 epoch. The proposed SNN training is significantly more carbon emitting compared to alternative of DNN/BNNs.*
> > > > >
> > > > > Response: These are again, we assert, misleading statements! **Our one-time-step SNNs do not take significantly more training epochs to converge. In fact, they are similar to standard DNNs (~300-400 epochs for VGG16 on CIFAR10). On the other hand, BNNs are significantly more training-expensive, and hence cost more carbon to train compared to our one-time-step SNNs due to their iterative pre-training, as evident from your reference [1] and the related Github repository readme.**
> > > > >
> > > > > Comment: *As with 1 time step the results on DVS is significantly poor, and as the accuracy is of equivalent BNNs are significantly high, this again demotivates the use of the proposed work compared to BNN alternatives, as now it should be applied to DVS to replace 1 step SNN. This further weakens the contribution of this work.*
> > > > >
> > > > > Response: **We again wonder how can someone compare the accuracy of BNNs, which lack any temporal dynamics, in DVS tasks!** The accuracy of one-time-step SNNs is poor (however better compared to SOTA approaches at one time step) compared to multi-time-step SNNs in DVS datasets, because they are enriched with significant temporal information, unlike static vision datasets. The fact **that our approach can be ported to DVS datasets, extended to multiple time steps, and trained to achieve SOTA accuracy, strengthens (and not weakens) the contribution of our work.**
> > > > >
> > > > > Comment: *Table 3 has comparisons of the SNN backbone with BNNs, models of different architecture. We would also like to highlight the ReActNet is a binary variant of ResNet18, that the authors compared with FP-32 weight SNN with ResNet50 backbone!!*
> > > > >
> > > > > Response: As mentioned earlier, our primary goal is not to compare our approach with bi-polar BNNs. We compare with them on object detection due to the lack of SNN and uni-polar BNN references for object detection tasks. Nevertheless, we have clearly mentioned the architecture we are comparing against, and so we are not sure why you claim our comparisons are ambiguous. These kinds of comparisons with architectural differences are fairly common in the vision community (as long as the architectural details are provided in the comparison), and it is not possible to evaluate one work on every architectural configuration! We are also happy to compare our work with any ResNet-50-based BNN used for object detection, but, we were unable to find any such model.
> > > > >
> > > > > Lastly, it seems like you believe in the openness of research and strongly advocate for a proper evaluation of our paper. We believe this is indeed in good spirits. In fact, we also believe that the open reviewing approach of ICLR is beneficial to our community. Hence, **we are open to discussing this with you, however, we are baffled by your irresponsible and misleading claims**.
> > > > >
> > > > > We have one last question for you, which we hope you will be able to answer. **Being a researcher affiliated with HKUST, how come you do not have an openreview account with an institutional email address or authentic webpage/LinkedIn?**
> > > > >
> > > > > References
> > > > >
> > > > > [1] Xu et al., ReCU: Reviving the Dead Weights in Binary Neural Networks, ICCV 2021.
> > > > >
> > > > > [2] Liu et al., ReActNet: Towards Precise Binary Neural Network with Generalized Activation Functions, ECCV 2020.
> > > > >
> > > > > [3] Tu et al., AdaBin: Improving Binary Neural Networks with Adaptive Binary Sets, ECCV 2022.
> > > > >
> > > > > [4] Chowdhury et al., Towards Ultra Low Latency Spiking Neural Networks for Vision and Sequential Tasks Using Temporal Pruning, ECCV 2022.
> > > > >
> > > > > [5] Yang et al., "DeepHoyer: Learning Sparser Neural Networks with Differentiable Scale-Invariant Sparsity Measures" ICLR 2020.

---

### Decision · Program_Chairs · 2023-01-20

**Decision:**

Reject

**Justification For Why Not Higher Score:**

The paper is not good enough for ICLR.

**Justification For Why Not Lower Score:**

N/A

**Metareview: Summary, Strengths And Weaknesses:**

The paper proposes to use a Hoyer regularizer and Hoyer spike layer to improve the training of one-time-step SNNs. The paper receives a borderline score and goes through intensive discussions between the reviewers and the authors. It has also been discussed in a panel via a virtual meeting. The reviewers’ original concerns are summarized as follows:
- The position of the considered one-time-step SNN in the areas of SNN and BNN, as well as its significance and biological plausibility
- The limited novelty of applying Hoyer regularizer
- The clarity of the motivation and explanation for the Hoyer regularizer and Hoyer spike layer
- Some details of experiment results, missing comparisons and experiments

During the discussion period, the authors made revisions to explain the position of this work as a continuum between sparsity-induced BNNs and SNNs for static vision tasks, emphasize the novelty of Hoyer spike layer, and supplement additional experiment results to better explain the method and demonstrate extensions. The reviewers acknowledged that some concerns are addressed, but still had concerns about the novelty, the explanation for the method, the bio-plausibility and the network capacity, as well as the clarity of experimental settings and hardware consideration.

During the panel discussion, the reviewers acknowledged the good empirical results, while raising concerns in two main aspects. First, about hardware consideration and energy efficiency. One-time-step SNN is more like ANN with binary activation rather than SNN. If training one-time-step SNN does not target for neuromorphic chips with bio-plausibility, it should compare with BNN more on common hardware such as GPU, which may hardly utilize the sparsity. Current results do not show advantages over BNN in a consistent evaluation. Second, about explanation for the method. The inconsistency of the Hoyer regularizer and Hoyer spike layer on unnormalized and normalized membrane potentials with trainable threshold is not explained well. Applying the Hoyer regularizer is not theoretically novel, so the reasonable explanation for the Hoyer spike layer is important.

Given the comments and discussions, the current version of the paper needs much additional revision to fully address concerns.

So based on the majority of negative rating from reviewers, the AC recommended rejection and all the reviewers (including qbCE) agreed on this decision during the virtual meeting. The AC appreciated the authors’ great efforts and hoped that the authors could use the constructive feedback to improve the paper. BTW, the AC wanted to point out that Eqn. (1) and (10) are incorrect.

**Summary Of Ac-Reviewer Meeting:**

Please see the above.